# FROM GNNs TO TREES: MULTI-GRANULAR INTERPRETABILITY FOR GRAPH NEURAL NETWORKS

**Jie Yang[1], Yuwen Wang[1], Kaixuan Chen[2,3], Tongya Zheng[2,4],**
**Yihe Zhou[1], Zhenbang Xiao[1], Ji Cao[1], Mingli Song[2,3], Shunyu Liu[5,\*]**

[1]Zhejiang University,
[2]State Key Laboratory of Blockchain and Data Security, Zhejiang University,
[3]Hangzhou High-Tech Zone (Binjiang) Institute of Blockchain and Data Security,
[4]Big Graph Center, Hangzhou City University,
[5]Nanyang Technological University
{yang_jie,yuwenwang,chenkx,zhouyihe,xiaozhb,caoji2001,brooksong}
@zju.edu.cn, doujiang_zheng@163.com, shunyu.liu@ntu.edu.sg

## ABSTRACT

Interpretable Graph Neural Networks (GNNs) aim to reveal the underlying reasoning behind model predictions, attributing their decisions to specific subgraphs that are informative. However, existing subgraph-based interpretable methods suffer from an overemphasis on local structure, potentially overlooking long-range dependencies within the entire graphs. Although recent efforts that rely on graph coarsening have proven beneficial for global interpretability, they inevitably reduce the graphs to a fixed granularity. Such an inflexible way can only capture graph connectivity at a specific level, whereas real-world graph tasks often exhibit relationships at varying granularities (*e.g.*, relevant interactions in proteins span from functional groups, to amino acids, and up to protein domains). In this paper, we introduce a novel Tree-like Interpretable Framework (TIF) for graph classification, where plain GNNs are transformed into hierarchical trees, with each level featuring coarsened graphs of different granularity as tree nodes. Specifically, TIF iteratively adopts a graph coarsening module to compress original graphs (*i.e.*, root nodes of trees) into increasingly coarser ones (*i.e.*, child nodes of trees), while preserving diversity among tree nodes within different branches through a dedicated graph perturbation module. Finally, we propose an adaptive routing module to identify the most informative root-to-leaf paths, providing not only the final prediction but also the multi-granular interpretability for the decision-making process. Extensive experiments on the graph classification benchmarks with both synthetic and real-world datasets demonstrate the superiority of TIF in interpretability, while also delivering a competitive prediction performance akin to the state-of-the-art counterparts. Our code will be made publicly available at https://github.com/dutyj2020/TIF.

## 1 INTRODUCTION

Graphs, as ubiquitous structures, are extensively employed to represent complex relationships in various fields, such as social networks (Bu & Shin, 2023; Tian & Zafarani, 2024), biological systems (Caufield et al., 2023; Garg, 2024), and transportation networks (Rahmani et al., 2023; Xu et al., 2022). To effectively model the connectivity patterns inherent in graphs, Graph Neural Networks (GNNs) have demonstrated extraordinary capabilities, enabling significant advancements in a variety of graph-based downstream tasks (Levie et al., 2018; You et al., 2020; Vrček et al., 2023). However, despite their effectiveness, a key challenge remains in the interpretability of GNNs, as their complex mechanisms often act as "black boxes" (Yuan et al., 2022; Li et al., 2022b). This lack of transparency makes it difficult to understand and trust their inner decision-making processes, which is essential for many security-critical applications (Zhao & Barati, 2023; El-Dawy et al., 2024).

---

*Corresponding author.

To alleviate this issue, interpretable GNNs have emerged as a promising paradigm from research communities, aiming to identify and elucidate the role of specific subgraphs in shaping the decisions made by the models (Ming et al., 2019; Chen et al., 2022; Yin et al., 2023; Tygesen et al., 2023; Lan et al., 2024), as depicted in Figure 1(a). However,

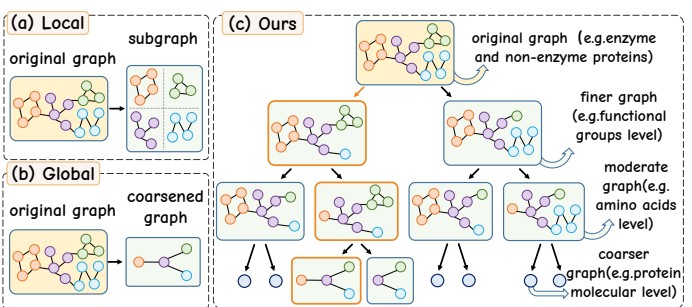

Figure 1: Comparing different interpretable methods for GNNs.

these subgraph-based interpretable methods often prioritize local structure at the expense of global context, potentially neglecting long-range dependencies within the entire graphs. Recent advancements highlight global interactions' importance for graph-level tasks, significantly improving GNNs' representation capacity (Yao et al., 2022; Ding et al., 2023; Zhili et al., 2024; Li et al., 2023; Liu et al., 2023). In light of this, a very recent work, GIP (Wang et al., 2024), introduces a learnable graph coarsening mechanism for global interpretability, as depicted in Figure 1(b). GIP aligns the coarsened graph instance with various self-interpretable graph prototypes, unveiling the model reasoning process from a global perspective. However, this global-based interpretable method inevitably reduces the graphs to a fixed granularity, capturing graph connectivity solely at the output level, thus failing to account for multi-granularity at intermediate levels. The fixed granularity may obscure intricate details in the inherent graph structure necessary for effective interpretability. Additionally, such an inflexible way can limit the model's adaptability to different graph types with diverse sizes and structures, thus compromising its robustness across various applications.

Graph-based tasks in real-world scenarios often involve relationships at multiple levels of granularity (Stawiski et al., 2000; Fan et al., 2019). For instance, the distinction between enzyme and non-enzyme proteins can be attributed to structural differences observed across varying granularities, ranging from *functional groups* and *amino acids* to *protein molecular* level (Hu et al., 2024; Zhai et al., 2024). At the level of *functional groups*, the functional groups in enzymes are organized into active sites with specific structures and orientations to facilitate catalysis, while non-enzymes lack such organized formations. When considered at the *amino acid* level, enzymes of the same type typically exhibit highly conserved amino acid sequences, ensuring consistency in structure and function across different instances of the enzyme. At the *protein molecular* level, enzymes often exhibit fewer helices and more elongated loops compared to non-enzymes, while also demonstrating tighter packing of their secondary structures (Stawiski et al., 2000). Thus, we suggest that interpretable GNNs can benefit from employing a multi-granular perspective. By spanning from local to global perspectives, interpretable GNNs can elucidate the underlying factors influencing model predictions across different levels of granularity, thereby enhancing trustworthiness for decision-makers.

In this paper, we introduce a new tree-like interpretable framework for graph classification, termed TIF, to explicitly transform original GNNs into hierarchical trees with each level representing coarsened graphs of varying granularities as tree nodes, as depicted in Figure 1(c). This tree-like structure is essential for multi-granular interpretability, as it can layer different granularities while using branches to capture diverse structural variations. TIF comprises three key modules that enable efficient tree construction and search. In the tree construction phase, TIF iteratively uses the graph coarsening module to reduce the original graphs (serving as root nodes of trees) into increasingly coarser graphs (serving as child nodes of trees), adding depth to the tree structure. Next, the graph perturbation module introduces learnable perturbations to ensure diversity among the tree nodes (*i.e.*, coarsened graphs) across different branches, which broadens the tree structure. In the tree search phase, the adaptive routing module dynamically identifies the most informative root-to-leaf paths, providing both the final prediction and multi-granular interpretability of the decision-making process. Our main contributions can be summarized as follows:

- We investigate a new challenge of multi-granular interpretability in GNNs, a highly important ingredient for graph-level tasks yet largely overlooked by existing literature.

- We propose a novel Tree-like Interpretable Framework (TIF) to transform plain GNNs into interpretable trees, thereby facilitating multi-granular interpretability. TIF employs the graph coarsening module and the graph perturbation module to build the tree structure, focusing on depth and breadth aspects respectively. Then the adaptive routing module is responsible for highlighting the valuable root-to-leaf paths for both model prediction and interpretability.

- Extensive experiments conducted on both synthetic and real-world datasets demonstrate that TIF yields competitive performance compared to state-of-the-art competitors, while significantly enhancing interpretability through multi-granular insights.

## 2 RELATED WORK

### 2.1 INTERPRETABLE GRAPH NEURAL NETWORKS

Traditional interpretable GNNs aim to uncover and explain the contribution of specific subgraphs to model decisions. These subgraph-based interpretable methods can be categorized into two main types: post-hoc methods and intrinsic methods. Post-hoc methods (Chen et al., 2022; Zhong et al., 2023; Fang et al., 2023), such as GNNExplainer (Ying et al., 2019) and PGExplainer (Luo et al., 2020), aim to explain model decisions by retrospectively finding key subgraphs after training. Intrinsic methods (Yin et al., 2023; Tilli & Vu, 2024; La Rosa, 2024), like GIB (Ming et al., 2019), incorporate explainability into the training process by emphasizing salient subgraphs. However, these subgraph-based methods tend to focus on local structures, often failing to capture long-range dependencies or global graph-level interactions (Ding et al., 2023).

Recent progress in GNN research has placed growing emphasis on the significance of global interactions in graph-level tasks (Yao et al., 2022; Ding et al., 2023; Zhili et al., 2024; Li et al., 2023; Liu et al., 2023). To address the limitations of subgraph-based interpretable methods in capturing global interactions, GIP (Wang et al., 2024) seeks to provide global interpretability by capturing interactions across the entire graph during training. While this global-based method improves global interpretability, it is often constrained by output level, limiting its ability to capture multi-scale relationships that are crucial for complex graph structures (Yao et al., 2022).

### 2.2 NEURAL TREES

Neural Trees (NTs) are the result of integrating Neural Networks (NNs) and Decision Trees (DTs). NTs can be classified into non-hybrid, semi-hybrid, and hybrid (Li et al., 2022a). Non-hybrid approaches extracted rules from trained NNs, but the two models were not integrated into a hybrid model (Costa & Pedreira, 2023; Ferigo et al., 2023; Bechler-Speicher et al., 2024; Costa et al., 2024). Semi-hybrid methods draw on the class hierarchy from DTs and incorporate it into NNs, but do not adopt the decision branch mechanism (Li et al., 2022a). Hybrid methods, or Neural Decision Trees (NDTs) (Zheng et al., 2023; Aissa et al., 2024), combine class hierarchy and decision branches (Li et al., 2022a), offering advantages in interpretability (Ji et al., 2020; Wan et al., 2021; Nauta et al., 2021). DNDF (Kontschieder et al., 2015) optimizes leaf predictions by minimizing a convex objective, but its explanations are derived solely from the final layer of the CNNs. Tanno et al. (2019) proposes dynamic tree generation to introduce interpretability into each layer of neural networks, but its greedy algorithms can result in suboptimal structures (Tanno et al., 2019). NBDT (Wan et al., 2021) uses predefined concepts to prevent suboptimal structures but relies on manual WordNet (Brust & Denzler, 2019) data. These works have developed quite richly in CNNs but are largely underexplored in GNNs. To address these limitations, we propose the TIF, enabling multi-granular and comprehensive interpretability in GNNs.

## 3 METHODOLOGY

In this section, we elaborate on the details of the proposed TIF. Figure 2 illustrates its workflow, which includes three modules: 1) The *hierarchical graph coarsening module* progressively compress the original graphs into coarser ones; 2) The *learnable graph perturbation module* introduces controlled perturbation matrices to preserve diversity among branches; 3) The *adaptive routing module* identifies the most informative root-to-leaf paths to make prediction and provide the multi-

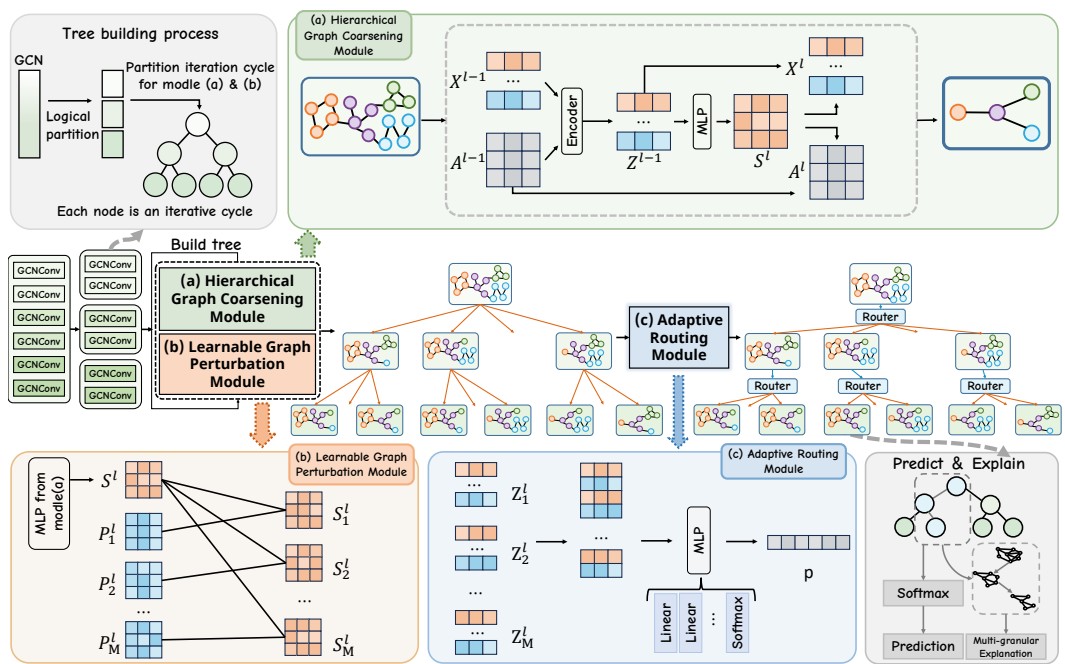

Figure 2: The workflow of the proposed TIF framework.

granular interpretability for decision-making. A detailed explanation of the formula notation in this chapter can be found in Appendix A.4.

## 3.1 HIERARCHICAL GRAPH COARSENING MODULE

In this section, we adopt a classical hierarchical graph pooling strategy (Ying et al., 2018; Duval & Malliaros, 2022; Islam et al., 2023; Wang et al., 2024)to build a tree-like interpretable framework with our dedicated perturbation and routing design in the next section. By iteratively applying this module, we capture multi-granular structural information.

Specifically, we first extract the node embeddings $\mathbf{Z}^{(l)}$ at each layer using Graph Convolutional Networks (GCNs) (Kipf & Welling, 2016). The update rule for the node embeddings is given by:

$$\mathbf{Z}^{(l)} = \sigma\left(\hat{\mathbf{D}}^{-\frac{1}{2}}\hat{\mathbf{A}}\hat{\mathbf{D}}^{-\frac{1}{2}}\mathbf{Z}^{(l-1)}\mathbf{W}^{(l)}\right), \tag{1}$$

where $\mathbf{Z}^{(0)} = \mathbf{X}$ denotes the input feature matrix, $\hat{\mathbf{A}} = \mathbf{A}+\mathbf{I}$ is the adjacency matrix with self-loops, $\hat{\mathbf{D}}$ is the degree matrix of $\hat{\mathbf{A}}$, $\mathbf{W}^{(l)}$ is the weight matrix, and $\sigma(\cdot)$ is the activation function. Then, we implement a *soft coarsening strategy* by using a multi-layer perceptron (MLP) with softmax on the output layer to generate a clustering assignment matrix $\mathbf{S}^{(l)}$ (Wang et al., 2024):

$$\mathbf{S}^{(l)} = \text{softmax}\left(\text{MLP}^{(l)}\left(\mathbf{Z}^{(l)}; \Theta_{\text{MLP}}\right)\right), \tag{2}$$

where $\Theta_{\text{MLP}}$ denotes trainable parameters, and $\mathbf{S}^{(l)}_{ij}$ represents the probability that node $v_i$ belongs to cluster $j$. Subsequently, we use the $\mathbf{S}^{(l)}$ to generate a new adjacency matrix $\mathbf{A}^{(l+1)}$ and a new embedding matrix $\mathbf{X}^{(l+1)}$ for coarser graph (Ying et al., 2018). We apply the following equations:

$$\mathbf{X}^{(l+1)} = \sum_{i=1}^{N} \mathbf{S}^{(l)\top}_{ji}\mathbf{Z}^{(l)}_i, \quad \forall j = 1, \ldots, K^{(l)}, \tag{3}$$

$$\mathbf{A}^{(l+1)} = \sum_{i=1}^{N}\sum_{k=1}^{N} \mathbf{S}^{(l)\top}_{ji}\mathbf{A}^{(l)}_{ik}\mathbf{S}^{(l)}_{kj}, \quad \forall j = 1, \ldots, K^{(l)}, \tag{4}$$

where $N$ is the number of nodes in the current layer, and $K^{(l)}$ is the cluster count at layer $l$. To preserve graph connectivity during coarsening, we apply *edge prediction loss* to constrain the process:

$$\mathcal{L}_{\text{link}} = -\sum_{i,j} \left( \mathbf{A}_{ij} \log \hat{\mathbf{A}}_{ij} + (1 - \mathbf{A}_{ij}) \log(1 - \hat{\mathbf{A}}_{ij}) \right), \tag{5}$$

where $\mathbf{A}_{ij}$ is the original adjacency matrix, and $\hat{\mathbf{A}}_{ij}$ is the adjacency matrix after coarsening.

## 3.2 LEARNABLE GRAPH PERTURBATION MODULE

In this module, we introduce multiple learnable perturbations for the coarsening process of each parent node in the tree, enhancing its representation diversity and robustness (Ying et al., 2019; Luo et al., 2020; Yuan et al., 2022). These slight perturbations allow similar child nodes to fully capture their similarity and share semantic paths across similar graphs, which not only facilitates the aggregation of structural information but also enriches the diversity of the tree, thus promoting the training of interpretable tree models.

Taking the expansion process of the $k$-th node in the $l$-th layer of the tree as an example, we first define $M$ learnable perturbation matrices for this node, *i.e.*, $\mathcal{P}^{l,k} = \{\mathbf{P}^{(l),k(1)}, \mathbf{P}^{(l),k(2)}, ..., \mathbf{P}^{(l),k(M)}\}$. Then, we use these perturbation matrices to perturb the clustering assignment matrix $\mathbf{S}^{(l),k}$ obtained by the graph coarsening module, which can be defined as:

$$\mathbf{S}^{(l),k(i)} = \mathbf{S}^{(l),k} + \mathbf{P}^{(l),k(i)}, \quad i = 1, 2, \ldots, M, \tag{6}$$

where $\mathbf{S}^{(l),k}$ represents the original clustering assignment matrix for the $k$-th node in the $l$-th layer of the tree. The perturbed node embeddings $\mathbf{X}^{(l),k(i)}$ will be computed based on the perturbed assignment matrices, which can be formulated as follows:

$$\mathbf{X}^{(l),k(i)} = \mathbf{S}^{(l),k(i)\top} \mathbf{Z}^{(l),k} = \mathbf{S}^{(l),k\top} \mathbf{Z}^{(l),k} + \mathbf{P}^{(l),k(i)\top} \mathbf{Z}^{(l),k}, \tag{7}$$

where $\mathbf{Z}^{(l),k}$ is the node embedding matrix for $k$-th parent node expansion at level $l$ of the tree. To ensure that the perturbed embeddings remain both useful and diverse, we introduce two regularization terms: *similarity regularization* and *diversity regularization*. The *similarity regularization* ensures that each perturbed embedding $\mathbf{X}^{(l),i}$ remains close to the original embedding $\mathbf{X}^{(l)}$, preserving important graph structure while applying perturbations:

$$\mathcal{L}_{\text{similarity}} = \sum_{l=1}^{L} \sum_{k=1}^{K^{(l)}} \sum_{i=1}^{M} \lambda_i \|\mathbf{X}^{(l),k(i)} - \mathbf{X}^{(l),k}\|^2, \tag{8}$$

where $\lambda_i$ controls the strength of the similarity term, $M$, $K^{(l)}$ and $L$ respectively represent the number of branches of the parent node, the number of parent nodes in each layer, and the number of layers in the tree. Additionally, we add the *diversity regularization* to promote variation between the perturbed embeddings, ensuring that each branch represents a different variant of the original graph:

$$\mathcal{L}_{\text{diversity}} = \sum_{l=1}^{L} \sum_{k=1}^{K^{(l)}} \mu \sum_{i \neq j} \|\mathbf{X}^{(l),k(i)} - \mathbf{X}^{(l),k(j)}\|^2, \tag{9}$$

where $\mu$ controls the degree of diversity.

In summary, these two terms form the total regularization loss $L_{\text{perturb}}$, which balances the preservation of the core graph structure with the need for diversity across branches at different levels:

$$\mathcal{L}_{\text{perturb}} = \mathcal{L}_{\text{similarity}} + \mathcal{L}_{\text{diversity}}. \tag{10}$$

## 3.3 ADAPTIVE ROUTING MODULE

In this module, we assign routers to each non-leaf node at every layer of the tree-like model for dynamically selecting the most informative root-to-leaf paths in the hierarchical structure.

First, we assign a router to each non-leaf node $k$ of layer $l$, which concatenates the perturbed embeddings $\{\mathbf{Z}^{(l),k(1)}, \mathbf{Z}^{(l),k(2)}, \ldots, \mathbf{Z}^{(l),k(M)}\}$ generated by the *learnable graph perturbation module* as inputs:

$$\hat{\mathbf{Z}}^{(l),k} = \text{MLP}([\mathbf{Z}^{(l),k(1)}; \mathbf{Z}^{(l),k(2)}; \ldots; \mathbf{Z}^{(l),k(M)}]). \tag{11}$$

Next, the router generates a set of routing logits $\mathbf{r}^{(l),k}$ based on the final node embedding $\hat{\mathbf{Z}}^{(l),k}$, which is used to represent the likelihood of choosing each path:

$$\mathbf{r}^{(l),k} = \mathbf{W}^{(2),r,k} \cdot \sigma(\mathbf{W}^{(1),r,k} \cdot \hat{\mathbf{Z}}^{(l),k} + \mathbf{b}^{(1),r,k}) + \mathbf{b}^{(2),r,k}, \qquad (12)$$

where $\mathbf{W}^{(1),r,k}$ and $\mathbf{W}^{(2),r,k}$ are the weight matrices for parent node $k$, $\mathbf{b}^{(1),r,k}$ and $\mathbf{b}^{(2),r,k}$ are bias terms, and $\sigma$ is a nonlinear activation function (*e.g.*, ReLU). The routing logits are then passed through a softmax function to yield the probability distribution $\mathbf{p}^{(l),k,i}$, representing the probability of choosing each path for parent node $k$ at layer $l$: $\mathbf{p}^{(l),k,i} = \text{softmax}(\mathbf{r}^{(l),k})^i$,. Based on the probability distribution, the most probable path $\hat{i}^{l,k}$ is selected: $\hat{i}^{l,k} = \arg\max_i \mathbf{p}^{(l),k,i}$. The node embedding and adjacency matrix of the next layer $l+1$ are updated accordingly based on the selected path:

$$\mathbf{X}^{(l+1),\hat{i}^{l,k}}_{\text{pooled}} = \mathbf{S}^{(l),\hat{i}^{l,k}\top}\mathbf{Z}^{(l)}, \quad \mathbf{A}^{(l+1),\hat{i}^{l,k}}_{\text{pooled}} = \mathbf{S}^{(l),\hat{i}^{l,k}\top}\mathbf{A}^{(l)}\mathbf{S}^{(l),\hat{i}^{l,k}}. \qquad (13)$$

This process is repeated across all layers, progressively coarsening the graph and refining path selection until the final node embeddings $\mathbf{X}^{(L+1)}$ and adjacency matrix $\mathbf{A}^{(L+1)}$ are obtained. To encourage the exploration of multiple paths, we introduce an entropy-based regularization, which promotes diversity in the path selection process:

$$\mathcal{L}_{\text{entropy}} = -\sum_{l=1}^{L}\sum_{k=1}^{K^{(l)}}\sum_{i=1}^{M}\mathbf{p}^{(l),k,i}\log(\mathbf{p}^{(l),k,i}), \qquad (14)$$

where $\mathbf{p}^{(l),k,i}$ is the probability assigned to each path for parent node $k$ at layer $l$.

## 3.4 INTERPRETABLE CLASSIFICATION BASED ON NEURAL TREE STRUCTURES

In this module, we make graph classification based on the constructed hierarchical tree-like model, which integrates multi-granularity information from different levels of the tree by tracking the identified root-to-leaf path, thus providing accurate and interpretable decisions.

For each test graph $\mathcal{G}_t$, we calculate the probability of path selection $\text{Path}^{(l),k}$ at each layer of the tree:

$$p(\text{Path}^{(l),k} \mid \mathcal{G}_t) = \frac{\exp(f(\text{Path}^{(l),k}, \mathcal{G}_t))}{\sum_j \exp(f(\text{Path}^{(l),j}, \mathcal{G}_t))}, \qquad (15)$$

where $f(\text{Path}^{(l),k}, \mathcal{G}_t)$ is a scoring function that measures the relevance of path $k$ at layer $l$ for the classification of graph $\mathcal{G}_t$. Then, we choose the path with the highest probability from all the options:

$$\hat{k}^{(l)} = \arg\max_k p(\text{Path}^{(l),k} \mid \mathcal{G}_t). \qquad (16)$$

This process is repeated iteratively for each layer until reaching the leaf node, resulting in a sequence of paths $\{\hat{k}^1, \hat{k}^2, \ldots, \hat{k}^L\}$ to denote the multi-granularity interpretation results of our method, where $L$ is the total number of layers in the tree.

The embeddings from the last selected path, $\mathbf{Z}^{\hat{k}^L}$, is directly used as the final embedding: $\hat{\mathbf{Z}} = \mathbf{Z}^{\hat{k}^L}$, which will be passed through a scoring function $f(\cdot)$ along with softmax to obtain the probability distribution for classification: $\mathbf{h}_i = \text{softmax}(f(\hat{\mathbf{Z}}))$, where $\mathbf{h}_i$ represents the model's predicted probabilities for assigning graph $G_i$ to each class. To ensure the accuracy of the proposed framework, we use the cross-entropy loss as the optimization objective, which can be denoted as:

$$\mathcal{L}_{\text{CE}} = \frac{1}{M}\sum_{i=1}^{M}\text{CrsEnt}(\mathbf{h}_i, \mathbf{y}_i) \qquad (17)$$

where $M$ is the batch size, $\mathbf{y}_i$ is the true probability distribution.

In summary, the final loss function combines the classification loss, edge prediction loss, entropy regularization, and perturbation regularization:

$$\mathcal{L}_{\text{total}} = \mathcal{L}_{\text{CE}} + \alpha_1 \mathcal{L}_{\text{link}} + \alpha_2 \mathcal{L}_{\text{perturb}} + \alpha_3 \mathcal{L}_{\text{entropy}}, \qquad (18)$$

where $\alpha_1$, $\alpha_2$, and $\alpha_3$ control the strength of edge prediction regularization, perturbation regularization, and entropy regularization, respectively.

## 4 EXPERIMENTS

### 4.1 EXPERIMENTAL SETTINGS

#### 4.1.1 DATASETS

We first conduct experiments on five real-world datasets across different domains to evaluate the effectiveness of our framework. Additionally, we use three synthetic datasets to better demonstrate the interpretability of our framework. The specifics of the datasets are as follows:

- **Real-world datasets:** To explore the effectiveness of our framework across different domains, we use protein datasets including ENZYMES, PROTEINS (Feragen et al., 2013), and D&D (Dobson & Doig, 2003), molecular dataset MUTAG (Wu et al., 2018), and scientific collaboration dataset COLLAB (Yanardag & Vishwanathan, 2015). Please refer to Appendix A.1 for more details.
- **Synthetic datasets:** We further adopt manually constructed datasets: GraphCycle, Graph-Five (Wang et al., 2024), and the MultipleCycle dataset we designed to better illustrate the interpretability of our framework. These datasets are composed based on the combination of structures at certain granularity levels, thus possessing a multi-level granular information structure. Graph-Cycle consists of two classes: Cycle and Non-Cycle, while GraphFive comprises five classes: Wheel, Grid, Tree, Ladder, and Star. MultipleCycle consists of four classes: Pure Cycle, Pure Chain, Hybrid Cycle, and Hybrid Chain. Implementation details are provided in Appendix A.1.

#### 4.1.2 BASELINES

We extensively compare our framework against three categories of baseline models:

- **Widely-used GNNs:** We evaluate prediction performance of TIF with several powerful GNNs, including GCN (Kipf & Welling, 2016), DGCNN (Wang et al., 2019), DiffPool (Ying et al., 2018), RWNN (Nikolentzos & Vazirgiannis, 2020), and GraphSAGE (Hamilton et al., 2017).
- **Subgraph-based Interpretable GNNs:** We compare the explanation performance with methods that adopt *post-hoc interpretation strategy*, including GNNExplainer (Ying et al., 2019), SubgraphX (Yuan et al., 2021), and XGNN (Yuan et al., 2020). We also compare both prediction and explanation performance with methods that adopt *intrinsic interpretation strategy*, such as Prot-GNN (Zhang et al., 2022), KerGNN (Feng et al., 2022), $\pi$-GNN (Yin et al., 2023), GIB (Yu et al., 2020), GSAT (Miao et al., 2022), and CAL (Sui et al., 2022).
- **Global-based Interpretable GNNs:** We also use GIP (Wang et al., 2024), which focuses on global interpretability, as a baseline to evaluate both the prediction performance and the explanation performance with our framework.
- **Neural Tree:** In addition, we construct a variant of TIF, which is a binary tree model with a single layer of linear routers, called Bi-Tree, for comparing the stability of the explanation.

More details about the baseline models and settings can be found in Appendix A.2 and A.3.

#### 4.1.3 METRICS

We follow previous work (Wang et al., 2024) to employ prediction and explanation performance for quantitative analysis. For *prediction performance*, we use **classification accuracy** and **F1 score** for evaluation. For *explanation performance*, considering that evaluating intrinsic explanations is non-trivial due to the lack of common evaluation criteria, we design four unique metrics:

- **Explanation Accuracy:** We use a trained GNN to predict the explanations generated by different methods and use the predicted confidence scores as a measure of explanation accuracy.
- **Consistency:** We adopt a random walk graph kernel to calculate the similarity between the explanations generated by different methods and the ground truth as a measure of consistency.
- **Path Consistency and Path Importance:** In order to compare the explanation stability with Bi-tree, we repeatedly input test samples into the model, record the path selection during each run, and calculate the consistency rate to measure path consistency. Additionally, we analyze the frequency of path utilization across a sample set, integrating gradient-based or feature contribution analyses, and using normalized entropy to measure the path importance.

## 4.2 QUANTITATIVE ANALYSIS

To validate the effectiveness of our framework, we first compare its performance in terms of prediction and interpretability against baseline models across several graph classification datasets.

### 4.2.1 PREDICTION PERFORMANCE

To validate the predictive performance of our approach, we compare our framework with widely used GNNs and interpretable GNN models on real-world and synthetic datasets. We apply three independent runs and represent the results in Table 1. We can draw the following observations:

- **Our framework matches or surpasses widely used GNN models in prediction performance.** Our framework improves accuracy by 0.09% to 35.77% on MUTAG and achieves top or second-best F1 scores in six of eight datasets, with comparable performance elsewhere.
- **Our framework outperforms existing interpretable GNN models significantly.** It achieves higher accuracy in six of eight datasets and higher F1 score in four.

Table 1: Comparison of different methods in terms of classification accuracy (%) and F1 score (%). We analyze the average results of three independent runs. **Bold** and underline denote the best and the second-best results, respectively. The results with std values can be found Appendix C.1.

| Method | ENZYMES | | D&D | | PROTEINS | | MUTAG | | COLLAB | | GraphCycle | | GraphFive | | MultipleCycle | |
|---|---|---|---|---|---|---|---|---|---|---|---|---|---|---|---|---|
| | Acc. | F1 | Acc. | F1 | Acc. | F1 | Acc. | F1 | Acc. | F1 | Acc. | F1 | Acc. | F1 | Acc. | F1 |
| GCN | 57.23 | 51.32 | 76.15 | 69.12 | 78.89 | 72.21 | 71.82 | 63.18 | 72.56 | 65.78 | 79.45 | 71.56 | 57.37 | 53.44 | 59.64 | 55.56 |
| DGCNN | 59.12 | 54.89 | 78.23 | 71.76 | 75.36 | 71.43 | 58.67 | 49.21 | 74.88 | 68.22 | 81.12 | 75.34 | 57.29 | 54.43 | 60.71 | 56.33 |
| Diffpool | 61.01 | 56.98 | 81.56 | 75.43 | 79.52 | **78.22** | 84.12 | 72.45 | 72.89 | **70.12** | 78.34 | 71.87 | 55.46 | 53.57 | 56.87 | 53.21 |
| RWNN | 54.76 | 48.12 | 76.89 | 74.67 | 76.12 | 70.89 | 88.21 | 85.04 | 73.45 | 68.45 | 78.89 | **78.76** | 56.25 | 52.45 | 57.16 | 54.09 |
| GraphSAGE | 58.12 | 44.89 | 79.34 | 79.23 | 79.04 | 68.45 | 74.23 | 71.78 | 71.23 | 65.45 | 77.45 | 72.12 | 59.11 | 52.72 | 62.66 | 59.34 |
| ProtGNN | 53.21 | 43.89 | 76.12 | 75.23 | 76.89 | 72.45 | 80.34 | 61.23 | 70.12 | 67.89 | 80.12 | 72.34 | 56.38 | 54.32 | 60.26 | 58.41 |
| KerGNN | 55.67 | 48.45 | 72.89 | 68.23 | 76.12 | 71.12 | 71.45 | 62.12 | 74.12 | 69.12 | 80.21 | 73.89 | 58.06 | 50.82 | 63.22 | 57.94 |
| $\pi$-GNN | 55.34 | 47.12 | 79.12 | 73.89 | 72.34 | 68.12 | 90.12 | 75.12 | 73.45 | 68.34 | 81.45 | 76.78 | 60.14 | 54.07 | 64.74 | 62.48 |
| GIB | 45.12 | 31.67 | 77.34 | 66.45 | 75.12 | 70.34 | 91.03 | 82.12 | 73.34 | 61.89 | 80.67 | 74.12 | 59.78 | **59.24** | 63.23 | 63.02 |
| GSAT | **61.34** | 55.12 | 72.12 | 67.12 | 74.45 | 71.89 | 94.35 | 82.34 | 75.87 | 63.78 | 80.12 | 75.08 | 59.58 | 54.13 | 66.49 | 65.24 |
| CAL | 61.12 | **58.12** | 78.12 | 68.78 | 74.56 | 67.12 | 89.78 | 85.12 | 77.12 | 64.12 | 81.42 | 78.12 | 56.49 | 50.93 | 61.77 | 58.94 |
| GIP | 60.61 | 57.41 | 79.32 | 75.78 | 79.55 | 75.28 | 91.21 | **86.73** | **77.49** | 67.47 | 82.15 | 78.31 | 60.38 | 54.98 | 68.72 | 66.45 |
| Ours | 58.66 | 55.44 | **84.19** | **81.01** | 79.96 | 77.21 | **94.44** | 86.23 | 77.29 | 67.82 | **84.77** | 78.49 | **64.35** | 55.07 | **69.04** | **67.91** |

### 4.2.2 EXPLANATION PERFORMANCE

We further compare our approach against subgraph-based and global-based interpretable methods in terms of explanation performance. We analyze the average results, and obtain four observations:

- **The accuracy of the explanations provided by our framework is competitive.** We compare our approach with subgraph-based and global-based interpretable methods, and the results are shown in Table 2. Compared to subgraph-based interpretable methods, our approach achieves the highest explanation accuracy on five of eight datasets and the second-best on the rest. Compared to only global-based interpretable baseline GIP, our approach also achieves comparably on most datasets.
- **Our framework can provide the most similar explanation to the ground-truth.** We compute the consistency between the explanations generated by different methods and the ground truth on two synthetic datasets, as shown in Figure 3(a). The consistency of our framework is significantly higher than most subgraph-based and global-based interpretable methods.
- **Our framework maintains relatively stable decision paths under varying conditions for the same input.** We compare path consistency between Bi-Tree and our model, the results are shown in Figure 3(b). It can be observed that our method achieves higher path selection consistency than the built explanation baseline of Bi-Tree on all datasets.
- **Our framework exhibits a well-balanced distribution of path importance across all datasets.** We conduct a comparative experiment on the path importance between Bi-Tree and our framework, the results are shown in Figure 3(c). It indicates that the path importance distribution of our method is balanced across all datasets, with no single path disproportionately dominating in importance. This balance suggests that the model does not overly rely on a few decision paths.

Table 2: Comparison of different methods in terms of explanation accuracy.

| Method | ENZYMES | D&D | PROTEINS | MUTAG | COLLAB | GraphCycle | GraphFive | MultipleCycle |
|---|---|---|---|---|---|---|---|---|
| ProtGNN | $85.12 \pm 2.12$ | $80.34 \pm 2.45$ | $69.12 \pm 3.12$ | $71.23 \pm 2.56$ | $80.67 \pm 2.78$ | $81.23 \pm 1.56$ | $72.12 \pm 1.34$ | $74.86 \pm 2.15$ |
| KerGNN | $63.45 \pm 2.45$ | $60.78 \pm 2.12$ | $79.12 \pm 1.45$ | $88.12 \pm 0.78$ | $84.11 \pm 1.12$ | $85.78 \pm 0.34$ | $75.67 \pm 0.67$ | $76.41 \pm 1.22$ |
| $\pi$-GNN | $75.12 \pm 1.12$ | $81.34 \pm 0.34$ | $65.45 \pm 2.67$ | $81.30 \pm 4.12$ | $76.12 \pm 0.34$ | $83.45 \pm 1.45$ | $64.89 \pm 0.34$ | $70.51 \pm 3.28$ |
| GIB | $72.89 \pm 2.12$ | $76.34 \pm 2.78$ | $82.78 \pm 1.67$ | $85.12 \pm 2.45$ | $78.45 \pm 2.34$ | $86.08 \pm 1.01$ | $79.07 \pm 0.56$ | $80.74 \pm 2.54$ |
| GSAT | $82.45 \pm 2.67$ | $75.12 \pm 0.45$ | $60.34 \pm 1.23$ | $75.34 \pm 3.56$ | $75.89 \pm 2.78$ | $90.12 \pm 2.34$ | $59.12 \pm 1.01$ | $68.38 \pm 1.79$ |
| CAL | $77.78 \pm 1.34$ | $74.12 \pm 3.12$ | $64.12 \pm 2.45$ | $76.12 \pm 1.37$ | $84.78 \pm 1.22$ | $84.12 \pm 2.12$ | $\mathbf{82.48 \pm 2.33}$ | $84.12 \pm 2.12$ |
| GNNExplainer | $80.12 \pm 0.45$ | $79.45 \pm 2.12$ | $\underline{87.12 \pm 2.45}$ | $82.13 \pm 2.12$ | $71.34 \pm 3.12$ | $85.41 \pm 2.78$ | $70.12 \pm 2.45$ | $77.34 \pm 2.05$ |
| SubgraphX | $81.67 \pm 2.45$ | $71.23 \pm 1.01$ | $75.89 \pm 2.12$ | $87.45 \pm 3.12$ | $76.34 \pm 3.12$ | $91.12 \pm 2.01$ | $69.50 \pm 3.45$ | $80.44 \pm 2.06$ |
| XGNN | $\mathbf{87.34 \pm 2.12}$ | $74.45 \pm 2.56$ | $74.12 \pm 2.01$ | $83.12 \pm 4.01$ | $84.45 \pm 0.56$ | $86.12 \pm 0.34$ | $76.45 \pm 1.17$ | $84.37 \pm 3.81$ |
| GIP | $86.08 \pm 2.60$ | $\underline{83.47 \pm 2.74}$ | $86.04 \pm 2.36$ | $\mathbf{90.05 \pm 1.44}$ | $\underline{85.21 \pm 3.72}$ | $92.79 \pm 1.32$ | $78.76 \pm 1.57$ | $\underline{85.16 \pm 2.68}$ |
| Ours | $\underline{86.53 \pm 2.01}$ | $\mathbf{89.11 \pm 1.26}$ | $\mathbf{87.62 \pm 2.12}$ | $88.21 \pm 1.34$ | $\mathbf{85.95 \pm 3.64}$ | $\mathbf{93.12 \pm 1.12}$ | $\underline{82.16 \pm 1.33}$ | $\mathbf{86.95 \pm 2.70}$ |

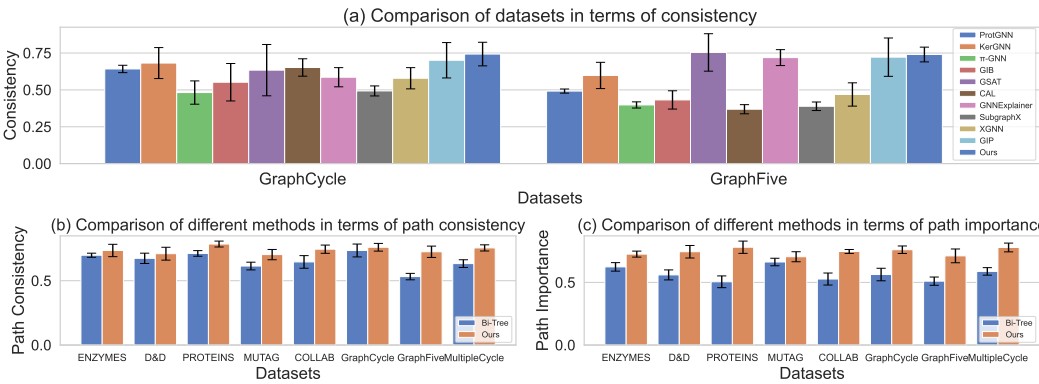

Figure 3: Explanation comparison on (a) consistency, (b) path consistency, (c) and path importance.

### 4.3 QUALITATIVE ANALYSIS

To comprehensively evaluate the interpretability of our proposed TIF, we conduct a detailed analysis of the multi-granular graph-level nodes and root-to-leaf paths it captures. To facilitate the observation of relationships between structures at different granularities, we visualize our framework's reasoning process for the MultipleCycle dataset and use different colors to distinguish between various substructures, as illustrated in Figure 4. We observe that TIF effectively captures both local substructures in finer explanations and global graph patterns in coarser explanations, ensuring that key features at different granularities are preserved. The adaptive routing module dynamically selects the most informative paths through the tree based on multi-granular complexity. We also process the MultipleCycle dataset using the GIP model and compare the explanations it generates with those produced by our Framework, as illustrated in Figure 5. Compared to GIP, TIF's capability to span from fine-grained local interactions to coarse-grained global structures provides a more transparent and interpretable decision-making process, elucidating how various levels of graph information contribute to final model predictions. More results of the explanation will be presented in Appendix B.

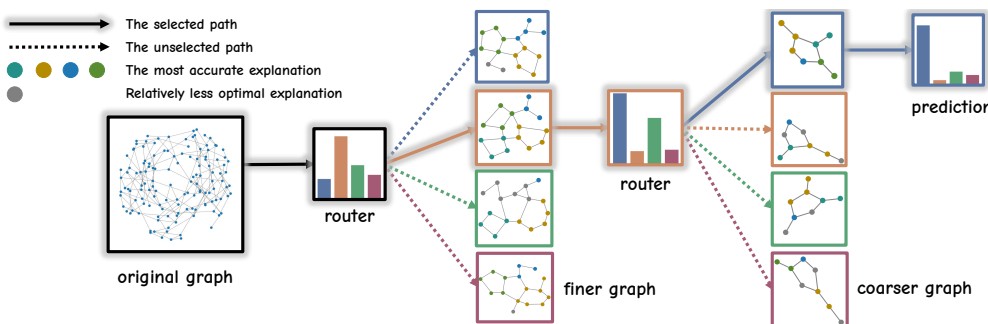

Figure 4: Explanations generated by our framework on MultipleCycle.

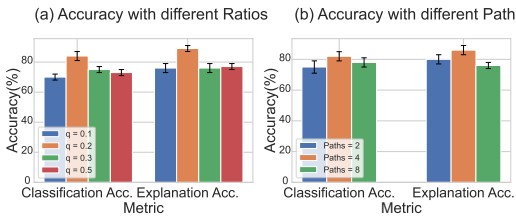

Figure 5: Explanation comparison generated by our framework and GIP on MultipleCycle.

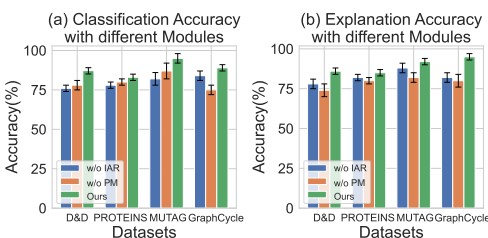

Figure 6: Performance w.r.t (a) compression ratios, (b) paths per node.

Figure 7: Performance w.r.t different modules.

### 4.4 ABLATION STUDIES

We conduct ablation studies to evaluate the impact of key components in our model: the compression ratio, the number of paths, and the routing complexity. More results will be shown in Appendix C.3.

First, we analyze the effect of the **compression ratio** $q$ (the ratio of nodes between layers). We vary $q$ across $\{0.1, 0.2, 0.3, 0.5\}$. We conduct experiments on the D&D dataset. Results in Figure 6(a) show that both classification and interpretability accuracy decrease when $q$ is too high or too low. A low $q$ retains noisy structures, while a high $q$ leads to loss of critical information.

Next, we investigate the **number of paths** $N$ by adjusting $N$ to $\{2, 4, 8\}$. We conduct experiments on the D&D. Results in Figure 6(b) show that the best classification performance was achieved with 4 paths while using fewer paths constrained information fusion and more paths introduced noise. Thus, 4 paths offer an effective balance between information use and noise control. Based on the explanation accuracy metric, it can be observed that the best interpretability was achieved with 4 paths while using fewer paths constrained information fusion and more paths introduced noise.

Finally, we examine **routing complexity** and **perturbation effect** by replacing the MLP-based routing module with a simpler linear structure(without the inter-layer adaptive routing mechanism, w/o IAR) and replacing the perturbation module(without the perturbation module, w/o PM). The results in Figure 7 show that TIF outperforms the other two variants in both classification and interpretability tasks. This suggests that TIF's routing structure better captures complex relationships between paths, while its perturbation structure effectively captures and learns information that benefits both classification tasks and interpretability.

## 5 CONCLUSION

In this paper, we propose the Tree-like Interpretable Framework (TIF), a novel approach for graph classification that introduces multi-granular interpretability by transforming GNNs into hierarchical trees. Unlike existing methods focused on local subgraph analysis or fixed granularity, TIF leverages iterative graph coarsening and perturbation mechanisms to capture diverse structural patterns across multiple granularities, ensuring a more comprehensive understanding of both global and local dependencies. The adaptive routing module dynamically selects the most informative root-to-leaf paths, improving both classification performance and interpretability. Extensive experiments on synthetic and real-world datasets demonstrate the superiority of TIF in providing competitive predictive performance while significantly enhancing the multi-granular interpretability of decision-making processes. By addressing the overlooked challenge of multi-granular interpretability, our work opens new avenues for the development of flexible, transparent, and robust graph neural networks in real-world applications. In the future, we will further explore scaling the interpretation framework from medium-scale graphs to large-scale graphs at moderate computational costs.

ACKNOWLEDGMENTS

This work was supported in part by the CCF-Baidu Open Fund under Grant No. CCF-BAIDU OF202410, in part by the Hangzhou Joint Funds of the Zhejiang Provincial Natural Science Foundation of China under Grant No. LHZSD24F020001, in part by the Zhejiang Province High-Level Talents Special Support Program "Leading Talent of Technological Innovation of Ten-Thousands Talents Program" under Grant No. 2022R52046, in part by the Zhejiang Provincial Natural Science Foundation of China under Grant No. LMS25F020012, and in part by the advanced computing resources provided by the Supercomputing Center of Hangzhou City University.

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

# A   MORE IMPLEMENTATION DETAILS

## A.1   DATASETS

The **ENZYMES** dataset, a collection of protein data obtained from the BRENDA database (Feragen et al., 2013), involves the classification of enzymes into one of six primary EC categories. Detailed statistics of this dataset are presented in Table 3.

The **PROTEINS** dataset, derived from the Dobson and Doig collection (Feragen et al., 2013), consists of protein data with the objective of distinguishing between enzymes and non-enzymes. Table 3 provides detailed statistics of this dataset.

The **D&D** dataset (Dobson & Doig, 2003) comprises high-resolution protein structures taken from a non-redundant selection of the Protein Data Bank. In this dataset, nodes represent amino acids, and an edge is formed between two nodes if they are less than 6 angstroms apart. Detailed statistics of the dataset can be found in Table 3.

The **MUTAG** dataset (Wu et al., 2018) is designed for predicting molecular properties, with nodes representing atoms and edges corresponding to chemical bonds. Each graph carries a binary label that indicates its mutagenic effect. Table 3 displays detailed statistics for the dataset.

The **COLLAB** dataset (Yanardag & Vishwanathan, 2015) focuses on scientific collaborations. In this dataset, each graph represents the ego network of a researcher, with nodes depicting the researcher and their collaborators, and edges signifying collaborations between researchers. The ego network of a researcher can be labeled with one of three categories: High Energy Physics, Condensed Matter Physics, or Astro Physics, reflecting the researcher's field of study. Detailed statistics of the dataset can be found in Table 3.

The **GraphCycle** dataset (Wang et al., 2024)is a synthetic dataset. Initially, 8~15 Barabási-Albert graphs are generated as communities, each with 10 to 200 nodes. These BA graphs are then interconnected to form two predefined shapes: Cycle and Non-Cycle. Edges between nodes in different communities are randomly added with a probability between 0.05 and 0.15. Detailed statistics of the dataset are given in Table 3.

The **GraphFive** dataset (Wang et al., 2024) is a synthetic dataset. Initially,8~15 Barabási-Albert graphs are generated as communities, each consisting of 10 to 200 nodes. These BA graphs are subsequently connected in five predefined shapes: Wheel, Grid, Tree, Ladder, and Star. To establish connections between nodes in different clusters, edges are randomly added with a probability between 0.05 and 0.15. Detailed statistics of the dataset can be found in Table 3.

**MultipleCycle** is a self-designed synthetic dataset. Specifically, we first generate random first-level structures, which consist of either a cycle or a non-cycle structure. For each node in this first-level structure, we further expand it by randomly generating second-level structures, which can either be a cycle or a non-cycle structure. Additionally, each node in the second-level structure is further expanded into one of four third-level structures: a triangle, star, trapezoid, or cycle. The dataset consists of four predefined categories: Pure Cycle, Pure Chain, Hybrid Cycle, and Hybrid Chain, determined based on whether the majority of the nodes at each level form cycle-based or chain-based structures. This hierarchical generation method ensures that each graph exhibits multiple levels of nested structures, with connectivity and patterns varying across the different classes. Specific statistics of the dataset are shown in Table 3.

Table 3: The statistics of real-world datasets.

|  | #Avg. Nodes | #Avg. Edges | #Classes | #Graphs |
|---|---|---|---|---|
| **ENZYMES** | 32.63 | 62.14 | 6 | 600 |
| **D&D** | 284.32 | 715.66 | 2 | 1178 |
| **PROTEINS** | 39.06 | 72.82 | 2 | 1113 |
| **MUTAG** | 17.93 | 19.79 | 2 | 188 |
| **COLLAB** | 74.49 | 2457.78 | 3 | 5000 |
| **GraphCycle** | 297.70 | 697.18 | 2 | 2000 |
| **GraphFive** | 375.98 | 1561.77 | 5 | 5000 |
| **MultipleGraph** | 175.33 | 263.41 | 4 | 5000 |

## A.2 BASELINE

To simplify the Tree-like Interpretable Framework (TIF) and investigate the impact of its core components on model performance, we designed a simplified model, named Bi-Tree.

### A.2.1 SIMPLIFIED LEARNABLE GRAPH PERTURBATION MODULE

In Bi-Tree, the learnable graph perturbation module from TIF has been simplified to use a set of fixed perturbation terms for each layer. Specifically, while TIF allows each parent node to have independent learnable perturbation matrices, Bi-Tree defines a set of fixed perturbation matrices $P_i^{(l)}$ for each layer $l$, corresponding to path $i$. The equation is as follows:

$$S_k^{(l)}(i) = S_k^{(l)} + P_i^{(l)}, \quad i = 1, 2, \ldots, M,$$
(19)

where $S_k^{(l)}$ represents the clustering assignment matrix generated by the graph coarsening module, and $P_i^{(l)}$ is the fixed perturbation matrix for path $i$ in layer $l$.

### A.2.2 BINARY TREE STRUCTURE WITH LINEAR ROUTERS

Bi-Tree constructs a binary tree structure, where each parent node has only two child nodes. Unlike TIF, which uses multi-level routers, Bi-Tree simplifies each layer's routers to linear transformations instead of multi-layer perceptrons (MLP). Specifically, the router computes the routing logits $r_k^{(l)}$ based on the node embeddings $Z_{\text{final},k}^{(l)}$:

$$r_k^{(l)} = W_{r,k} \cdot Z_{\text{final},k}^{(l)} + b_{r,k},$$
(20)

where $W_{r,k}$ is the weight matrix for parent node $k$, and $b_{r,k}$ is the bias term.

## A.3 HYPER-PARAMETER SETTINGS

The hyper-parameters used in our framework include batch size, optimizer, learning rate, and epoch. Additionally, several key hyper-parameters control the various loss terms in the model. Specifically, $\alpha_1$ controls the contribution of the edge prediction loss $\mathcal{L}_{\text{link}}$, which ensures the preservation of graph connectivity during the hierarchical graph coarsening process. $\alpha_2$ governs the perturbation regularization loss $\mathcal{L}_{\text{perturb}}$, balancing similarity regularization $\mathcal{L}_{\text{similarity}}$ and diversity regularization $\mathcal{L}_{\text{diversity}}$ to ensure the embeddings remain diverse yet close to the original during the learnable graph perturbation module. $\alpha_3$ adjusts the entropy regularization loss $\mathcal{L}_{\text{entropy}}$, which promotes diverse path selection in the adaptive routing module. The specific settings are provided in Table 4.

Table 4: The statistics of hyper-parameters setting.

|  | ENZYMES | PROTEINS | D&D | MUTAG | COLLAB | GraphCycle | GraphFive | MultipleGraph |
|---|---|---|---|---|---|---|---|---|
| **Batch Size** | 64 | 64 | 128 | 64 | 64 | 128 | 128 | 128 |
| **Optimizer** | Adam | Adam | Adam | Adam | Adam | Adam | Adam | Adam |
| **Learning Rate** | 0.001 | 0.003 | 0.001 | 0.001 | 0.003 | 0.01 | 0.01 | 0.01 |
| **Epoch** | 500 | 500 | 500 | 500 | 500 | 500 | 500 | 500 |
| $\alpha_1/\alpha_2$ | 0.3/0.2 | 0.3/0.2 | 0.3/0.2 | 0.3/0.2 | 0.3/0.2 | 0.3/0.2 | 0.3/0.2 | 0.3/0.2 |
| $\alpha_3$ | 0.1 | 0.1 | 0.1 | 0.1 | 0.1 | 0.1 | 0.1 | 0.1 |

## A.4 MORE DETAILED EXPLANATION OF THE NOTATION

To enhance the readability of the formulas, we will provide a symbol table to further elaborate on the specific meanings of each subscript, offering detailed explanations for each subscript and its function. This will particularly focus on how these subscripts are used in the tree structure model to represent different levels, nodes, and perturbation terms, helping readers better understand our notation system. For details, please refer to Table 5, 6, 7, 8, 9, 10 and 11.

Table 5: Node-related symbols.

| Symbol | Subscript/Superscript | Meaning and Role |
|---|---|---|
| $v_i$ | $i$ | The $i$-th node in the graph, representing a specific node. |
| $\mathbf{Z}^{(l)}$ | $l$ | Node embedding matrix after graph convolution at layer $l$, containing embeddings for all nodes. |
| $\mathbf{Z}^{(l),k}$ | $k,l$ | Node embeddings belonging to node $k$ at layer $l$, used for representing tree nodes. |
| $\mathbf{Z}^{(l),k(i)}$ | $k(i),l$ | Embeddings of node $k$ perturbed by the $i$-th perturbation at layer $l$. |
| $\hat{\mathbf{Z}}^{(l),k}$ | $k,l$ | Final aggregated embedding for node $k$ at layer $l$, used for routing and decisions. |

Table 6: Feature and weight-related symbols.

| Symbol | Subscript/Superscript | Meaning and Role |
|---|---|---|
| $\mathbf{X}$ | None | Input feature matrix, containing the original graph's node features. |
| $\mathbf{X}^{(l),k}$ | $k,l$ | Feature matrix of node $k$ at layer $l$, describing its feature state. |
| $\mathbf{X}^{(l),k(i)}$ | $k(i),l$ | Feature matrix of node $k$ after applying the $i$-th perturbation at layer $l$. |
| $\mathbf{X}^{(l+1)}$ | $l+1$ | Feature matrix of the coarsened graph at layer $l+1$. |
| $\mathbf{W}^{(l)}$ | $l$ | Weight matrix of the graph convolution at layer $l$, used for learning graph structural features. |
| $\mathbf{W}^{(1),r,k}, \mathbf{W}^{(2),r,k}$ | $r,k$ | Router weight matrices for node $k$ at layer $l$, used to compute path selection probabilities. |
| $\mathbf{b}^{(1),r,k}, \mathbf{b}^{(2),r,k}$ | $r,k$ | Bias terms for the router of node $k$ at layer $l$. |

Table 7: Graph structure-related symbols.

| Symbol | Subscript/Superscript | Meaning and Role |
|---|---|---|
| $\mathbf{A}$ | None | Adjacency matrix of the original graph, representing node connectivity. |
| $\hat{\mathbf{A}}$ | $\hat{\ }$ | Adjacency matrix with self-loops added, improving the stability of graph convolution operations. |
| $\mathbf{A}^{(l)}$ | $l$ | Adjacency matrix of the graph at layer $l$, describing node connectivity in the coarsened graph. |
| $\mathbf{A}^{(l+1),\hat{i}^{l,k}}_{\text{pooled}}$ | pooled, $\hat{i}^{l,k}, l+1$ | Adjacency matrix of the coarsened graph generated for the selected path $\hat{i}^{l,k}$. |

Table 8: Clustering-related symbols.

| Symbol | Subscript/Superscript | Meaning and Role |
|---|---|---|
| $\mathbf{S}^{(l)}$ | $l$ | Clustering assignment matrix at layer $l$, representing the probabilities of nodes belonging to different clusters. |
| $\mathbf{S}^{(l),k}$ | $k,l$ | Clustering assignment matrix for node $k$ at layer $l$. |
| $\mathbf{S}^{(l),k(i)}$ | $k(i),l$ | Clustering assignment matrix for node $k$ under the $i$-th perturbation at layer $l$. |

Table 9: Loss and regularization-related symbols.

| Symbol | Subscript/Superscript | Meaning and Role |
|---|---|---|
| $\mathcal{L}_{\text{link}}$ | link | Edge prediction loss, ensuring connectivity of the adjacency matrix during graph coarsening. |
| $\mathcal{L}_{\text{similarity}}$ | similarity | Similarity regularization, constraining perturbed embeddings to remain close to the original embeddings. |
| $\mathcal{L}_{\text{diversity}}$ | diversity | Diversity regularization, promoting differences between perturbed embeddings. |
| $\mathcal{L}_{\text{entropy}}$ | entropy | Entropy regularization, encouraging diversity in path selection. |
| $\mathcal{L}_{\text{CE}}$ | CE | Cross-entropy loss, optimizing classification objectives. |
| $\mathcal{L}_{\text{total}}$ | total | Total loss function, combining classification, edge prediction, perturbation, and entropy losses. |

Table 10: Path and routing-related symbols.

| Symbol | Subscript/Superscript | Meaning and Role |
|---|---|---|
| $\mathbf{r}^{(l),k}$ | $k,l$ | Routing logits for node $k$ at layer $l$, used to compute path selection probabilities. |
| $\mathbf{p}^{(l),k,i}$ | $k,i,l$ | Path selection probability for node $k$ at layer $l$, representing the likelihood of selecting branch $i$. |
| $\hat{i}^{l,k}$ | $l,k$ | Optimal path index for node $k$ at layer $l$, selected based on the maximum probability. |
| $\text{Path}^{(l),k}$ | $k,l$ | Path set at layer $l$, describing the paths associated with node $k$. |

Table 11: Parameters and hyperparameters.

| Symbol | Subscript/Superscript | Meaning and Role |
|---|---|---|
| $\lambda_i$ | $i$ | Weight of the similarity regularization term, controlling the strength of the $i$-th perturbation. |
| $\mu$ | None | Weight of the diversity regularization term, controlling variation between perturbations. |
| $\alpha_1, \alpha_2, \alpha_3$ | $1,2,3$ | Weight coefficients for edge prediction, perturbation, and entropy regularization terms, respectively. |
| $M$ | None | Number of perturbation branches for each node. |
| $N$ | None | Number of nodes in the current layer. |
| $K^{(l)}$ | $l$ | Number of clusters at layer $l$. |
| $L$ | None | Total number of layers in the tree. |

## B    Additional Visual Explanations

### B.1    Additional Visual Explanations for the Tree Structure

To comprehensively evaluate the interpretability of our proposed TIF, we provide an example that contains the input graph, the root-to-leaf path, the coarsened graphs of each layer, and the final prediction. We conduct a detailed analysis of the multi-granular graph-level nodes and root-to-leaf paths it captures. To facilitate the observation of relationships between structures at different granularities, we visualize our framework's reasoning process for the MultipleCycle dataset and use different colors to distinguish between various substructures, as illustrated in Figure 8. We observe that TIF effectively captures both local substructures in finer explanations and global structure in coarser explanations, ensuring that key features at different granularities are preserved. The routing module selects the most informative paths through the tree based on multi-granular complexity.

Below, we will take Figure 8 as an example and provide a detailed analysis of the entire process, starting from the input graph, progressing through each intermediate layer and the root-to-leaf path, and finally arriving at the output graph and prediction results and elaborate correlation between the coarsened graph at each layer and the ground-truth.

Firstly, the input graph is a sample from the MultipleCycle dataset, and its category is "Hybrid Cycle". It corresponds to different ground truths at different levels of granularity. Specifically:

- Its first-level structure is set as a cycle structure based on the ground truth at this granularity level, which determines its cycle attribute.

- Its second-level structure is built on the first-level structure, configured as a mixed combination of cycle and non-cycle structures according to the ground truth at this granularity level. The clockwise sequence is cycle, non-cycle, cycle, and cycle, which determines its mixed attribute. (for more detailed information on the dataset, please refer to Appendix A.1.)

Therefore, the final prediction for the input graph in this dataset requires the model to determine:

- whether its first-level granular structure is cycle or non-cycle.

- whether its second-level granular structure represents a mixed combination.

In other words, the model is expected to analyze and make determinations at different granularity levels for this dataset.

Secondly, when the input graph is fed into the model. After passing through a series of graph convolution layers and being processed by the Graph Perturbation Module and Routing Module at the root node of the TIF, the model produces four finer graphs.

We can observe that the finer graphs clearly display the second-level structure of the input graph (in the figures, different colors are used to annotate the nodes of the finer graphs, distinguishing the various second-level structures). From left to right:

- The first finer graph shows a second-level structure starting from the top-left and proceeding clockwise as cycle, non-cycle, cycle, and non-cycle (this structure is not clearly represented).

- The second finer graph shows a second-level structure proceeding clockwise as non-cycle, cycle (which is somewhat ambiguous and not purely cycle), cycle, and cycle.

- The third finer graph shows clockwise as cycle, non-cycle, cycle, and cycle.

- The fourth finer graph shows clockwise as cycle, non-cycle, cycle, and non-cycle.

The model selects the third finer graph, which best reflects the structural information of the input graph. From an interpretability perspective, this layer of finer graphs in the TIF tree model captures the second-level structural information of the input graph. Furthermore, the model selects the finer graph that most effectively represents the second-level structure of the input graph (clockwise: cycle, non-cycle, cycle, cycle). From the perspective of ground truth, the model selects the finer graph that is closest to the ground truth structure and layout of the input graph at this granularity level.

Subsequently, the selected finer graph undergoes another series of graph convolution layers and is processed by the Learnable Graph Perturbation Module and the Adaptive Routing Module at the next layer of the TIF. The model then produces four coarser graphs.

We can observe that the coarser graphs clearly capture the first-level structure of the input graph, which is the cycle structure. To illustrate this correspondence, we have used different colors in the figures to annotate the nodes of the coarser graphs, aligning them with the structures of the finer graphs from the previous layer. From left to right:

- The first coarser graph has two nodes extending outward as small structures from the cycle.
- The second coarser graph has three discontinuous nodes extending outward from the cycle.
- The third coarser graph has two nodes extending outward as small structures from the cycle.
- The fourth coarser graph has three nodes extending outward as small structures from the cycle structure, corresponding to the second-level structure depicted in the finer graph from the previous layer (three cycles organized consecutively).

The model selects the fourth coarser graph, which best represents the structural information of the input graph, as the root node of the TIF. From an interpretability perspective, this layer of coarser graphs in the TIF captures the first-level structural information of the input graph. Additionally, the model selects the coarser graph that not only most effectively represents the first-level structural information of the input graph but also retains the second-level structural information (clockwise: cycle, non-cycle, cycle, cycle, i.e., three cycles organized consecutively). From the perspective of ground truth, the model selects the finer graph that is closest to the ground truth structure and layout of the input graph at this granularity level, while also most accurately preserving the ground truth structural information from the previous granularity level.

Finally, at the root node of the TIF, the prediction is performed, and the model successfully identifies the data as "Hybrid Cycle". From an interpretability perspective, the TIF effectively captures and explains the key attributes of the MultipleCycle dataset at two distinct granularity levels.

- The second-level granularity characterizes the attributes of being purely cycle, purely non-cycle, or a mixed combination of cycle and non-cycle structures.
- The first-level granularity identifies whether the structure is cycle or non-cycle.

Based on these attributes at the two different granularity levels, the model successfully makes the final prediction for the input graph, completing the classification task.

In addition, the relationship between each coarsened graph and the ground truth lies in the fact that each coarsened graph in the TIF strives to represent the critical structures constructed by the ground truth at the granularity level that the layer aims to explain for the input graph. That is, the coarsened graph obtained at each level by TIF corresponds to the ground truth at that level of granularity.

### B.2    ADDITIONAL VISUAL EXPLANATIONS ON DIFFERENT DATASETS

In this section, we will present additional visualization outcomes of explanations on different datasets. We visualize the explanations generated by our framework on the PROTEINS and D&D datasets. The outcomes are presented in Figure 9 and Figure 10. For clarity of presentation, we only show partial sections of the full explanations for the *finer graph* granularity and *moderate graph* granularity. It can be easily observed that TIF effectively captures both local substructures and global graph patterns, ensuring that key features at different granularities are preserved.

For example, in the PROTEINS dataset, compared to the explanations for non-enzymes, the explanations for enzymes at the *protein molecular* level, or the *coarser graph* granularity, display more long loops and tighter connections. At the *amino acid* level, or the *moderate graph* granularity, enzyme explanations show relatively fixed structural combinations. At the *functional group* level, or the *finer graph* granularity, enzyme explanations reveal denser connections at the active sites.

This observation offers us new insights into differentiating graphs with varying properties, even without specialized knowledge. In the future, we plan to collaborate with domain experts to perform a more thorough analysis.

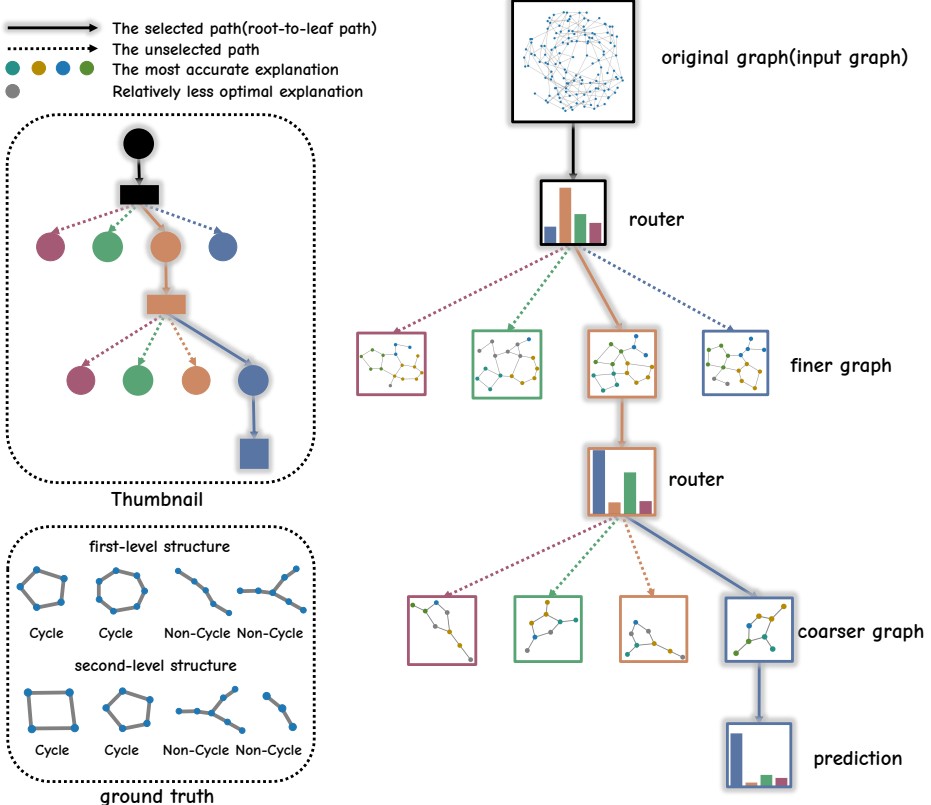

Figure 8: An example which contains the input graph, the root-to-leaf path, the coarsened graphs of each layer, and the final prediction.

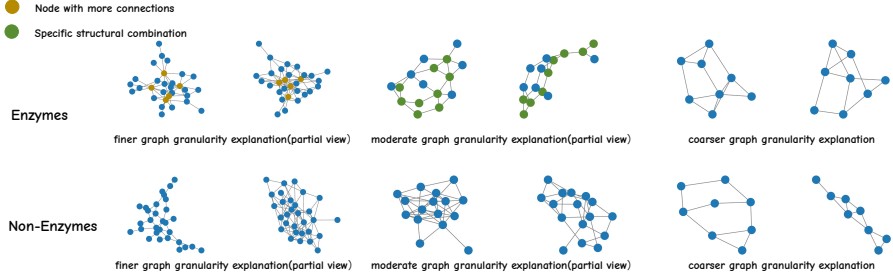

Figure 9: Explanations generated by our framework on the PROTEINS dataset.

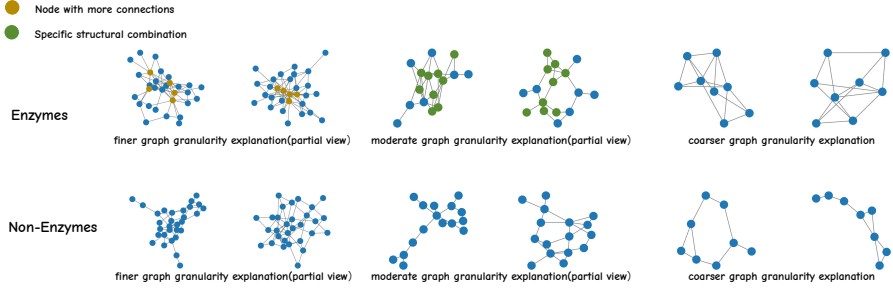

Figure 10: Explanations generated by our framework on the D&D dataset.

### B.3 ADDITIONAL VISUAL EXPLANATIONS ON DIFFERENT METHODS

In this section, we observe that TIF effectively captures both local substructures in finer explanations and global graph patterns in coarser explanations, ensuring that key features at different granularities are preserved. The adaptive routing module dynamically selects the most informative paths through the tree based on multi-granular complexity. We also process the same samples using the GIP, GSAT, and ProtGNN and compare the explanations it generates with those produced by our Framework, as illustrated in Figure 11. Our standard for explaining quality is the ability to accurately capture the important features and structural information at each granularity level. Different colors represent structural information learned or captured from the previous level of granularity. Therefore, models like GIP only provide a template based on the entire graph, so the generated explanation is depicted in gray. Compared to those models, TIF's capability to span from fine-grained local interactions to coarse-grained global structures provides a more transparent and interpretable decision-making process, elucidating how various levels of graph information contribute to final model predictions.

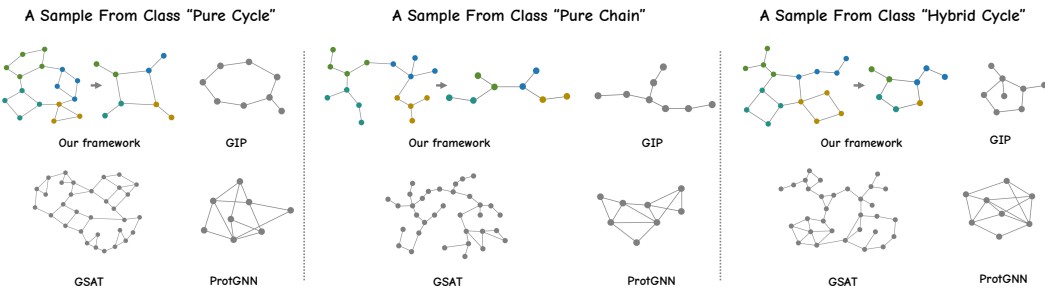

Figure 11: Explanation comparison generated by TIF, GIP, GSAT and ProtGNN on MultipleCycle.

## C   MORE DETAILED EXPERIMENTAL RESULTS

### C.1   PREDICTION PERFORMANCE WITH STD VALUE

To validate the predictive performance of our approach, we compare our framework with widely used GNNs and interpretable GNN models on real-world and synthetic datasets. We apply three independent runs and report the results along with their corresponding std values in Table 12.

Table 12: Comparison of different methods in terms of classification accuracy (%) and F1 score (%) along with their corresponding standard deviations.

| Method | ENZYMES | | D&D | | PROTEINS | | MUTAG | | COLLAB | | GraphCycle | | GraphFive | | MultipleCycle | |
|---|---|---|---|---|---|---|---|---|---|---|---|---|---|---|---|---|
| | Acc. | F1 | Acc. | F1 | Acc. | F1 | Acc. | F1 | Acc. | F1 | Acc. | F1 | Acc. | F1 | Acc. | F1 |
| GCN | 57.23±0.81 | 51.32±0.33 | 76.15±2.77 | 69.12±1.02 | 78.89±0.90 | 72.21±3.14 | 71.82±4.27 | 63.18±4.36 | 72.56±1.09 | 65.78±3.75 | 79.45±1.04 | 71.56±1.02 | 57.37±0.81 | 53.44±0.55 | 59.64±4.70 | 55.56±3.34 |
| DGCNN | 59.12±3.30 | 54.89±1.88 | 78.23±0.78 | 71.76±1.59 | 75.36±1.92 | 71.43±3.53 | 58.67±0.80 | 49.21±0.42 | 74.88±2.38 | 68.22±1.44 | 81.12±2.72 | 75.34±3.11 | 57.29±3.33 | 54.43±2.87 | 60.71±1.07 | 56.33±2.41 |
| Diffpool | 61.01±2.26 | 56.98±2.55 | 81.56±1.31 | 75.43±4.68 | 79.52±0.78 | **78.22±0.87** | 84.12±2.18 | 72.45±2.60 | 72.89±1.39 | **70.12±1.17** | 78.34±4.23 | 71.87±4.59 | 55.46±1.21 | 53.57±1.57 | 56.87±2.03 | 53.21±2.22 |
| RWNN | 54.76±1.43 | 48.12±3.22 | 76.89±1.99 | 74.67±2.18 | 76.12±1.36 | 70.89±1.40 | 88.21±0.21 | 85.04±0.41 | 73.45±1.52 | 68.45±1.97 | 78.89±1.48 | **78.76±2.53** | 56.25±0.42 | 52.45±1.22 | 57.16±5.56 | 54.09±4.10 |
| GraphSAGE | 58.12±1.22 | 44.89±1.32 | 79.34±5.31 | _79.23±6.77_ | 79.04±2.15 | 68.45±2.08 | 74.23±3.27 | 71.78±3.62 | 71.23±1.58 | 65.45±2.51 | 77.45±1.49 | 72.12±2.47 | 59.11±0.34 | 52.72±0.36 | 62.66±0.01 | 59.34±0.77 |
| ProtGNN | 53.21±1.57 | 43.89±2.36 | 76.12±1.21 | 75.23±2.49 | 76.89±0.52 | 72.45±1.87 | 80.34±2.45 | 61.23±3.83 | 70.12±0.97 | 67.89±1.04 | 80.12±1.21 | 72.34±2.04 | 56.38±4.21 | 54.32±4.37 | 60.26±3.38 | 58.41±3.67 |
| KerGNN | 55.67±4.22 | 48.45±2.03 | 72.89±1.48 | 68.23±2.36 | 76.12±2.30 | 71.12±2.10 | 71.45±1.08 | 62.12±1.22 | 74.12±1.66 | 69.12±1.97 | 80.21±0.72 | 73.89±0.68 | 58.06±0.11 | 50.82±1.02 | 63.22±0.05 | 57.94±0.33 |
| π-GNN | 55.34±0.88 | 47.12±0.76 | 79.12±1.10 | 73.89±1.85 | 72.34±3.77 | 68.12±2.21 | 90.12±0.43 | 75.12±2.09 | 73.45±1.52 | 68.34±3.05 | 81.45±2.22 | 76.78±5.62 | 60.14±0.05 | 54.07±0.31 | 64.74±1.21 | 62.48±1.97 |
| GIB | 45.12±3.22 | 31.67±1.73 | 77.34±1.69 | 66.45±0.90 | 75.12±6.34 | 70.34±1.05 | 91.03±4.88 | 82.12±1.26 | 73.34±1.79 | 61.89±1.65 | 80.67±1.74 | 74.12±1.98 | 59.78±0.15 | **59.24±0.17** | 63.23±2.63 | 63.02±2.70 |
| GSAT | **61.34±0.65** | 55.12±1.47 | 72.12±1.13 | 67.12±3.22 | 74.45±0.79 | 71.89±1.48 | _94.35±1.12_ | 82.34±1.93 | 75.87±3.56 | 63.78±2.59 | 80.12±0.14 | 75.08±0.57 | 59.58±3.09 | 54.13±2.70 | 66.49±1.50 | 65.24±1.53 |
| CAL | 61.12±3.24 | **58.12±4.44** | 78.12±2.88 | 68.78±4.76 | 74.56±4.09 | 67.12±4.21 | 89.78±6.99 | 85.12±8.31 | 77.12±4.78 | 64.12±6.25 | 81.42±2.33 | 78.12±2.40 | 56.49±1.44 | 50.93±2.59 | 61.77±0.42 | 58.94±1.73 |
| GIP | 60.61±2.41 | _57.41±2.80_ | 79.32±1.01 | 75.78±0.36 | _79.55±0.61_ | 75.28±0.90 | 91.21±2.25 | **86.73±2.92** | 77.49±4.26 | 67.47±2.11 | _82.15±1.38_ | 78.31±2.66 | _60.38±3.33_ | 54.98±1.52 | _68.72±0.02_ | 66.45±1.34 |
| Ours | 58.66±1.44 | 55.44±2.50 | **84.19±0.88** | **81.01±0.76** | 79.96±0.97 | _77.21±0.34_ | **94.44±2.44** | _86.23±3.52_ | 77.29±2.08 | 67.82±3.27 | **84.77±0.92** | _78.49±1.16_ | **64.35±3.55** | _55.07±2.87_ | **69.04±0.21** | **67.91±2.77** |

### C.2   EFFICIENCY STUDY

In this section, we analyze the efficiency of the proposed TIF framework and compare its efficiency with several interpretable baselines.

The modular design of TIF ensures efficient computation by progressively reducing the number of nodes through hierarchical coarsening, while controlled perturbations and adaptive routing maintain computational feasibility without compromising model diversity and interpretability.

The running efficiency of the proposed TIF framework is analyzed as follows. In Table 13, we present the time required to complete the training of each interpretable model. The dataset is divided into 10 equal subsets for 10-fold cross-validation, with the time taken by each model being the average of the times required for each fold. Specifically, in each iteration, one fold is held out as the validation set, while the remaining 9 folds are used for training. It should be noted that $\pi$-GNN requires an additional pre-training process that takes nearly 72 hours, which significantly impacts its overall computational efficiency. Therefore, the efficiency of $\pi$-GNN is considerably lower than our framework. It can be seen that our framework is only slightly less efficient than the KerGNN model and GIP model. Given that our model outperforms KerGNN and GIP in terms of prediction and explanation performance on the vast majority of datasets, as analyzed above, we believe that this slight additional time cost is justified.

Table 13: Time consumption of different methods. The table shows the time required (in seconds) to finish training for each interpretable model on various datasets. "*" indicates the method requires an additional pre-training process which takes nearly 72 hours.

| Methods | ENZYMES | D&D | COLLAB | MUTAG | GraphCycle | GraphFive |
|---|---|---|---|---|---|---|
| ProtGNN | 10245.65s | 19312.87s | 38021.49s | 9239.15s | 14396.76s | 5022.81s |
| KerGNN | 384.73s | 1313.59s | 1927.34s | 401.34s | 198.45s | 458.22s |
| $\pi$-GNN* | 406.18s | 966.94s | 1747.55s | 462.94s | 283.74s | 429.82s |
| GIB | 711.57s | 2923.67s | 4681.74s | 3107.31s | 1159.82s | 1208.78s |
| GSAT | 482.61s | 1388.45s | 2979.63s | 828.19s | 568.27s | 649.34s |
| GIP | 437.51s | 1134.20s | 2008.77s | 452.26s | 235.67s | 423.87s |
| Ours | 433.17s | 1109.70s | 2251.30s | 503.18s | 359.69s | 488.15s |

## C.3 ADDITIONAL ABLATION STUDIES

### C.3.1 IMPACT OF THE COMPRESSION RATIO

In this section, we extend the analysis on the impact of the compression ratio $q$ on model performance, conducting experiments across datasets such as MUTAG, and PROTEINS. The results are presented in Figure 12 and Figure 13.

As discussed in the main text, we observe that both classification accuracy and interpretability accuracy tend to decline when the compression ratio is either too high or too low. Specifically, a low compression ratio may introduce noisy structures, thereby hindering the extraction of global information, while a high compression ratio might lead to the loss of critical information.

### C.3.2 IMPACT OF THE NUMBER OF PATHS

In this section, we present further results on the impact of the number of paths on model performance, covering datasets such as ENZYMES, COLLAB, and FiveGraph shown in Figure 14.

Consistent with the observations in the main text, the experiments reveal that the model achieves the best interpretability when the number of paths is set to four, while performance deteriorates when the number of paths is either too few or too many. Specifically, with only two paths, the model's choice space is constrained, resulting in insufficient information fusion and an inability to fully leverage the diversity of the graph structure. Conversely, when the number of paths is increased to eight, although potential information channels are expanded, additional noise is introduced, making it challenging for the model to focus on the most critical features. Thus, setting the number of paths to four strikes a balance between information utilization and noise control, effectively improving the model's interpretability and stability.

### C.3.3 IMPACT OF DIFFERENT MECHANISMS

In this section, we further examine routing complexity and perturbation effect by replacing the MLP-based routing module with a simpler linear structure(without the inter-layer adaptive routing mechanism, w/o IAR) and replacing the perturbation module(without the perturbation module, w/o PM).

Experiments were conducted across various datasets such as ENZYMES, COLLAB, and GraphFive, with results for classification accuracy and interpretability accuracy presented in Figure 15.

As shown in the figure, the experimental results indicate that the performance is slightly inferior when these mechanisms are used individually, while the combination of these mechanisms achieves the best performance. This superiority stems from the fact that the combination of these mechanisms helps to identify common characteristics in the graph from the perspective of global structure interactions, thereby effectively enhancing the model's ability to extract global information and interpret key features in complex graph structures. Specifically, the hierarchical graph coarsening module iteratively aggregates components with similar features or close connections at each layer, forming graph-level representations with higher levels of abstraction. Meanwhile, the graph perturbation module integrates learnable perturbation mechanisms within each lateral layer, resulting in graph-level representations that better reflect the hierarchical structure's layer-wise characteristics. The combination of these mechanisms is crucial for improving the overall performance of the model.

### C.3.4 IMPACT OF LEARNABLE GRAPH PERTURBATION MODULE

In this section, we analyze the impact of the Learnable Graph Perturbation Module on the model and its effectiveness in enhancing diversity. Based on TIF, we created two variants. The first variant replaces the original perturbation terms for each parent node with a set of learnable perturbation terms shared across all parent nodes in each layer(simplified version, SV). The second variant degrades the model by removing the branching structure entirely, effectively eliminating the Learnable Graph Perturbation Module(without the perturbation module, w/o PM).

Experiments were conducted across various datasets, with results for classification accuracy and interpretability accuracy presented in Figure 16.

As shown in the figure, the experimental results indicate that TIF outperforms the other two variants in both classification and interpretability tasks. This suggests that TIF's perturbation structure effectively captures and learns information that benefits both classification tasks and interpretability.

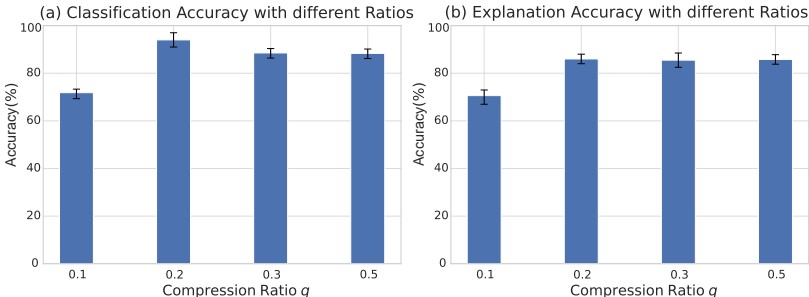

Figure 12: The influence of different compression ratios on the model on the MUTAG dataset.

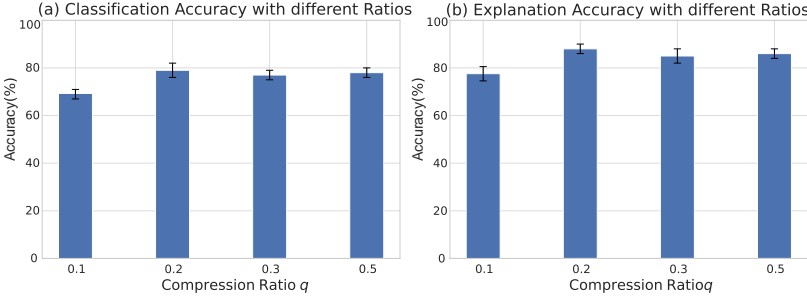

Figure 13: The influence of different compression ratios on the model on the PROTEINS dataset.

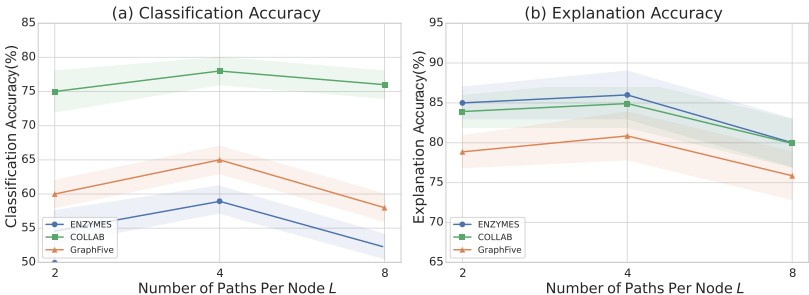

Figure 14: The influence of different numbers of paths per node on the model's effectiveness.

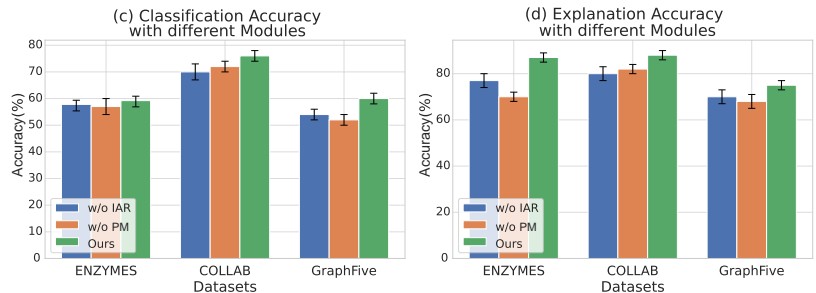

Figure 15: The influence of different modules on the model's effectiveness.

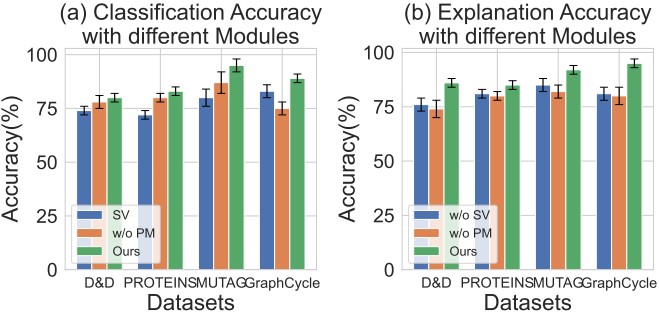

Figure 16: The influence of the learnable graph perturbation module on the model's effectiveness.

