# OpenReview forum: "From GNNs to Trees: Multi-Granular Interpretability for Graph Neural Networks"
_ICLR.cc/2025/Conference — ICLR 2025 Poster_

### Official Review · Reviewer_hGDZ · 2024-10-18

**Soundness:** 3
**Presentation:** 3
**Contribution:** 3
**Rating:** 6
**Confidence:** 5

**Summary:**

The authors propose a novel interpretable graph learning method, i.e., TIF, which transforms the GNN into a hierarchical tree. Within the hierarchical tree, each layer represents a graph granularity and each node represents a coarsened graph. In detail, the hierarchical graph coarsening module clusters the graph nodes into coarser clusters and then the learnable graph perturbation module introduce various perturbations to propel the robustness and the diversity of the coarsened graph. At last, an adaptive router identifies the best root-to-leaf path and provides the prediction result. Sufficient experiments demonstrate the effectiveness of TIF in both interpretation and prediction.

**Strengths:**

1. The tree structure based interpretation method is intriguing and can further enlighten the GNN explanation field.
2. The manuscript is well-organized and the proposed TIF framework is clearly introduced, making the whole work easy to understand.
3. The experiments are sufficient and discussed in detail, verifying the effectiveness of the proposed TIF model.
4. The Figures and Tables are insightful and well-analyzed, deriving meaningful conclusions that facilitates the understanding to TIF.

**Weaknesses:**

1. The definition of interpretation in this work is opaque. Whether the selected root-to-leaf path or the final leaf node serves as the interpretation to the input graph? Furthermore, why the coarsened graph can be considered as a kind of explanation? An example of input graph, the prediction result, and the corresponding explanation is helpful to figure this issue out.
2. The evaluation metrics adopted in experiments fail to reflect the sparsity of the generated explanation. A trivial explanation, such as the original graph, is meaningless to practical applications.
3. Some terminologies are used before giving definition. In Figure 3(c), the definition of path importance is missing in the manuscript. In Section 4.4, the compression ratio q is discussed without introduction or definition.
4. In Figure 4 and Figure 5, what is the principle to measure the quality of the generated explanations? For example, which instance of the finer graphs in Figure 4 is closer to the ground-truth explanation? Furthermore, the explanation provided by GIP is grey (without color), while that of TIF is colorful. If this difference represents some superiorities of TIF, and please elaborate if yes.
5. In Figure 7, the notations "IAR" and "LV" lack definition and the discussion of Figure 7 mismatches with it.

**Questions:**

1. Could you provide an example which contains the input graph, the root-to-leaf path, the coarsened graphs of each layer, and the final prediction. A detailed analysis to  the input, the explanation, and the prediction can facilitate the comprehension of TIF.
2. An ablation study towards the graph perturbation module is recommended, to demonstrate the effectiveness of enhancing diversity.

---

> ### Author Response · Authors · 2024-11-22
> **Response (1/2)**
>
> We are glad that the reviewer appreciates our work as a useful contribution to the community. We have carefully revised the manuscript according to your constructive suggestions. Below we address the main points raised in the review.
>
> >[W1] & [Q1] (1) Definition of Interpretation in This Work, (2) the Reason Why the Coarsened Graph Can Serve as an Interpretation, (3) an Example of the TIF Model.
>
> Thanks for the valuable comment. (1) In our work, the interpretability is defined from a multi-granularity perspective, including tree nodes with coarsened graphs of varying granularities, and the selected root-to-leaf path.
>
> (2) The coarsened graphs generated by our model can illustrate and reveal the structural features of the corresponding original graph data at a given granularity [1] (*as shown in Figures 4-5, 8-11 of the coarsened graph*). This provides structural explanations and learning guidance specific to the chosen granularity.
>
> (3) We now additionally provide an example in *Appendix B.1 (Page 16,18, Figure 8)*, which includes the input graph, the root-to-leaf path, the coarsened graphs of each layer, and the final prediction. Moreover, in the initial manuscript, we have also provided a similar example. *Figure 4 (Page 10)* demonstrates that the coarsened graphs deliver effective interpretability.
>
> We observe that TIF effectively captures both local substructures in finer explanations and global graph patterns in coarser explanations, ensuring that key features at different granularities are preserved. *These clarifications and additional experiments have been updated in Appendix B.1 (Page 16,18, Figure 8) of the revised manuscript.*
>
> >[W2] (1) The Experimental Metric That Reflects Sparsity, (2) Original Graph Is Meaningless to Practical Applications.
>
> Sorry for the confusion. (1) In our ablation studies of the initial manuscript (*Page 9-10, 20-22*), we have introduced the compression ratio as a sparsity metric to reflect the sparsity of the generated explanations. (2) The original graph is not included in the scope of our explanations. Instead, our generated explanations are based on coarsened graphs of different granularities (*Page 2, Figure 1 and line 72*).
>
> >[W3] Definition of (1) Path Importance and (2) Ratio q.
>
> (1) Sorry for the misunderstanding. Path Importance is defined as a measure to determine the significance of each path by analyzing its utilization frequency across a sample set and quantifying it using normalized entropy. This metric helps identify potential issues of imbalance, where certain paths are either excessively or insufficiently utilized. In the initial manuscript, we have provided the definition of Path Importance in *Section 4.1.3 METRICS on Page 7, line 374*.
>
> (2) Thanks for the valuable comment. The ratio q is defined as the measure of compression applied to the graph structure and node features between adjacent layers of the model during pooling. *These clarifications have been updated on Page 9, line 482 of the revised manuscript.*
>
> >[W4] Detailed Explanation of Figures 4 and 5.
>
> Thanks for the valuable comment. (1) The standard for explaining quality is the ability to accurately capture the important features and structural information at each granularity level. Different colors represent structural information learned or captured from the previous level of granularity. (2) For example, in Figure 4, the model selects a finer graph with three-ring structures and one chain structure, which corresponds to the structure of the original graph. Then, a coarser graph with three branches is selected, which also corresponds to the previously chosen finer graph's structure. (3) The comparison in Figure 5 mainly serves to demonstrate that TIF can provide multi-granularity explanations, whereas models like GIP cannot. These models can only recognize single structural information, such as rings or non-rings. Different colors represent the structural information learned or captured from the previous level of granularity. Therefore, models like GIP only provide a template based on the entire graph, so the generated explanation for GIP is depicted in gray. *These additional clarifications have been updated on Page 19, Appendix B.3.*
>
> >[W5] Notations in Figure 7.
>
> Sorry for the confusion. We have completed the missing symbol definitions and added a description of a specific model variant in Figure 7, ensuring that Figure 7 aligns with the corresponding discussion. *These clarifications have been updated in Figure 7 (Page 10) of the revised manuscript.*

---

> ### Author Response · Authors · 2024-11-22
> **Response (2/2)**
>
> >[Q2] Ablation Study for Perturbation Module.
>
> Thanks for the valuable comment. We have additionally designed two model variants for the Perturbation Module to conduct ablation experiments. Experiments were conducted across various datasets, with results for classification accuracy and interpretability accuracy presented in *Figure 16 (page 22)*. Moreover, in the initial manuscript, we have also provided a similar experiment (*Figure 7, Page 10* and *Figure 15, Page 22*). The experimental results indicate that TIF outperforms the other two variants in both classification and interpretability tasks. This suggests that TIF’s perturbation structure effectively captures and learns information that benefits both classification tasks and interoperability. *These additional experiments have been updated in Figure 16 (page 22) of the revised manuscript.*
>
>
>
> ---
>
> Ref:
>
> [1] Yuwen Wang, Shunyu Liu, Tongya Zheng, Kaixuan Chen, and Mingli Song. Unveiling Global Interactive Patterns across Graphs: Towards Interpretable Graph Neural Networks. In *SIGKDD (ACM Knowledge Discovery and Data Mining)*, pages 3277–3288, 2024.

---

> ### Author Response · Authors · 2024-11-25
> **Looking forward to your reevaluation**
>
> Dear reviewer,
>
> We are glad that the reviewer appreciates our attempt, and sincerely thank the reviewer for the constructive comments.
> As suggested, we have additionally added some clarifications about our framework and included further experiments, such as an example of the TIF model and an ablation study for the perturbation module. Please let us know if you have other questions or comments.
>
> Since half of the allocated time for reviewer-author discussion has already elapsed, we sincerely look forward to your reevaluation of our work and would very appreciate it if you could raise your score to boost our chance of more exposure to the community. Thank you very much!
>
> Best regards,
>
> The authors of TIF

---

> > ### Comment · Reviewer_hGDZ · 2024-11-25
> >
> > Dear authors,
> >
> > Thanks for your detailed responses which clarify most of my confusions.
> >
> > Nevertheless, the main concern towards the generated interpretation still exists. In Figure 8, since the input graph is selected from synthetic dataset MultipleCycle, the ground-truth interpretation ought to be available. Therefore, what the is correlation of the coarsen graph in each layer to the ground-truth interpretation? From my perspective, it remains obscure to understand the final prediction of the input graph, even if the root-to-leaf path and the coarsen graphs are provided.
> >
> > Best.

---

> > > ### Author Response · Authors · 2024-11-26
> > > **Response (1/2)**
> > >
> > > We are deeply honored by your response, and we sincerely appreciate your thoughtful feedback. Below, we will take Figure 8 as an example and provide a detailed analysis of the entire process, starting from the input graph, progressing through each intermediate layer and the root-to-leaf path, and finally arriving at the output graph and prediction results and elaborate correlation between the coarsened graph at each layer and the ground-truth. Additionally, We have added a visual structure of the ground truth used at each granularity level in the MultipleCycle dataset. *These additional clarifications have been updated on Page 18, Figure 8.*
> > > 1. input graph
> > > + Firstly, the **input graph** is a sample from the MultipleCycle dataset, and its category is classified as "Hybrid Cycle." It corresponds to different ground truths at different levels of granularity. Specifically:
> > >     - Its first-level structure is set as a cycle structure based on the ground truth at this granularity level, which determines its cycle attribute.
> > >     - Its second-level structure is built on the first-level structure, configured as a mixed combination of cycle and non-cycle structures according to the ground truth at this granularity level. The clockwise sequence is cycle, non-cycle, cycle, and cycle, which determines its mixed attribute. (for more detailed information on the dataset, please refer to Appendix A.1.)
> > > + Therefore, the final prediction for the input graph in this dataset requires the model to determine:
> > >     - whether its first-level granular structure is cycle or non-cycle.
> > >     - whether its second-level granular structure represents a mixed combination of cycle and non-cycle structures.
> > > + In other words, the model is expected to analyze and make determinations at different granularity levels for this dataset.
> > > 2. finer graph
> > > + Secondly, when the input graph is fed into the model. After passing through a series of graph convolution layers and being processed by the Graph Perturbation Module and Routing Module at the **root node** of the TIF, the model produces four **finer graphs**.
> > > + We can observe that the finer graphs clearly display the second-level structure of the input graph (in the figures, different colors are used to annotate the nodes of the finer graphs, distinguishing the various second-level structures). From left to right:
> > >     - The first finer graph shows a second-level structure starting from the top-left and proceeding clockwise as cycle, non-cycle, cycle, and non-cycle (this structure is not clearly represented).
> > >     - The second finer graph shows a second-level structure proceeding clockwise as non-cycle, cycle (which is somewhat ambiguous and not purely cycle), cycle, and cycle.
> > >     - The third finer graph shows a second-level structure proceeding clockwise as cycle, non-cycle, cycle, and cycle.
> > >     - The fourth finer graph shows a second-level structure proceeding clockwise as cycle, non-cycle, cycle, and non-cycle.
> > > + The model selects the third finer graph, which best reflects the structural information of the input graph. **From an interpretability perspective**, this layer of finer graphs in the TIF tree model captures the second-level structural information of the input graph. Furthermore, the model selects the finer graph that most effectively represents the second-level structure of the input graph (clockwise: cycle, non-cycle, cycle, cycle). **From the perspective of ground truth**, the model selects the finer graph that is closest to the ground truth structure and layout of the input graph at this granularity level.

---

> > > ### Author Response · Authors · 2024-11-26
> > > **Response (2/2)**
> > >
> > > 3. coarser graph
> > > + Subsequently, the selected finer graph undergoes another series of graph convolution layers and is processed by the Learnable Graph Perturbation Module and the Adaptive Routing Module at the next layer of the TIF. The model then produces four **coarser graphs**.
> > > + We can observe that the coarser graphs clearly capture the first-level structure of the input graph, which is the cycle structure. To illustrate this correspondence, we have used different colors in the figures to annotate the nodes of the coarser graphs, aligning them with the structures of the finer graphs from the previous layer. From left to right:
> > >     - The first coarser graph has two nodes extending outward as small structures from the cycle structure.
> > >     - The second coarser graph has three discontinuous nodes extending outward as small structures from the cycle structure.
> > >     - The third coarser graph has two nodes extending outward as small structures from the cycle structure.
> > >     - The fourth coarser graph has three nodes extending outward as small structures from the cycle structure, corresponding to the second-level structure depicted in the finer graph from the previous layer (three cycles organized consecutively).
> > > + The model selects the fourth coarser graph, which best represents the structural information of the input graph, as the root node of the TIF. **From an interpretability perspective**, this layer of coarser graphs in the TIF captures the first-level structural information of the input graph. Additionally, the model selects the coarser graph that not only most effectively represents the first-level structural information of the input graph but also retains the second-level structural information (clockwise: cycle, non-cycle, cycle, cycle, i.e., three cycles organized consecutively). **From the perspective of ground truth**, the model selects the finer graph that is closest to the ground truth structure and layout of the input graph at this granularity level, while also most accurately preserving the ground truth structural information from the previous granularity level.
> > > 4. prediction
> > > + Finally, at the **root node** of the TIF, the prediction is performed, and the model successfully identifies the category of the data as "Hybrid Cycle."
> > > + From an interpretability perspective, the TIF tree model effectively captures and explains the key attributes of the MultipleCycle dataset at two distinct granularity levels.
> > >     - The second-level granularity, which characterizes the attributes of being purely cycle, purely non-cycle, or a mixed combination of cycle and non-cycle structures.
> > >     - The first-level granularity, which identifies whether the structure is cycle or non-cycle.
> > > + Based on these attributes at the two different granularity levels, the model successfully makes the final prediction for the input graph, completing the classification task.
> > > 5. correlation of coarsen graph to the ground-truth
> > > + In addition, the relationship between each coarsened graph and the ground truth lies in the fact that each coarsened graph in the TIF model strives to represent the critical structures constructed by the ground truth at the granularity level that the layer aims to explain for the input graph. That is, the coarsened graph obtained at each level by TIF corresponds to the ground truth at that level of granularity.

---

> > > > ### Comment · Reviewer_hGDZ · 2024-11-26
> > > >
> > > > Dear authors,
> > > >
> > > > Thanks again for your detailed elaboration. I have raised my score to 6: marginally above the acceptance threshold and good luck.
> > > >
> > > > Best.

---

> > > > > ### Author Response · Authors · 2024-11-27
> > > > > **Thanks for your positive support**
> > > > >
> > > > > We are deeply grateful for your positive support and thoughtful encouragement. Have a nice day！

---

### Official Review · Reviewer_Yj5d · 2024-10-30

**Soundness:** 2
**Presentation:** 3
**Contribution:** 2
**Rating:** 6
**Confidence:** 3

**Summary:**

This paper introduces a Tree-like Interpretable Framework (TIF) for inerpretable graph classification, transforming standard GNNs into hierarchical trees that represent graphs at varying granularities. TIF uses a graph coarsening module to compress graphs while maintaining diversity among tree nodes, and an adaptive routing module to highlight the most informative paths for decision-making. Extensive experiments show that TIF improves interpretability and achieves competitive prediction performance compared to state-of-the-art methods.

**Strengths:**

1. It is reasonable to require hierarchical explanations for the graph classification tasks of GNNs, and the issue addressed in this paper is of practical significance.
2. The paper employs sufficient comparative algorithms to demonstrate the effectiveness of the proposed algorithm.
3. The introduction is well-written and provides clarity, greatly aiding in the understanding of the paper's contributions and innovations.

**Weaknesses:**

1. The notation in the methods section is quite confusing due to the various subscripts, making it very difficult to follow.
2. The evaluation metrics used to measure the effectiveness of model explanations are not sufficiently convincing.
3. There is a lack of visual comparisons between the various interpretable GNNs, such as GSAT vs the proposed method.
4. There is a lack of analysis regarding the algorithm's complexity and running efficiency.
5. The proposed algorithms lack clear theoretical foundations and appear to be based solely on heuristic designs.

**Questions:**

1. In Fig. 1, what do "finer graph" and "moderate graph" represent? Why is the progression from finer to moderate to coarsen? It is recommended to explain this in the caption or the main text.
2. what is the relationship between $ \hat A$ and $A^{(l+1)}$?  Additionally, as shown in eq (5), does the coarsened adjacency matrix $ \hat A$ have the same dimensions $n*n$ as the adjacency matrix of the original graph? Why eq(5) can help the model ensure the connectivity of the coarsened graph?
3. Regarding the explanation accuracy metric, can the softmax output from the GNN be considered as confidence? This lacks theoretical justification.
4. Regarding the consistency metric, is the capability of graph kernels sufficient for distinguishing structural differences?
5. Conducting explanation experiments on the Mutag dataset is necessary.
6. Please provide visual explanations of algorithms like GSAT on the MultipleCycle dataset.

---

> ### Author Response · Authors · 2024-11-22
> **Response (1/4)**
>
> We greatly appreciate the reviewer for the insightful comments, which helped improve the quality of the paper significantly. We have carefully revised the manuscript according to your valuable suggestions. Below we address the main points raised in the review.
>
> >[W1] Various Subscripts Making It Difficult to Follow.
>
> Sorry for the confusion. We have reorganized the formula expressions throughout the paper. Subscripts are consistently used to represent the internal logic of matrices and classification tasks, while superscripts are used to indicate the logic related to the tree model structure. Moreover, we provide a series of symbol tables to further elaborate on the specific meanings of each subscript. *These clarifications have been updated in Table 5-11 (Page 16-17) of the revised manuscript, as follows:*
>
> Table 5: Node-related Symbols.
>
> |**Symbol**|**Subscript/Superscript**|**Meaning and Role**|
> |-|-|-|
> |$v_i$|$i$|The $i$-th node in the graph, representing a specific node.|
> |$Z^{(l)}$|$l$|Node embedding matrix after graph convolution at layer $l$, containing embeddings for all nodes.|
> |$Z^{(l),k}$|$k, l$|Node embeddings belonging to node $k$ at layer $l$, used for representing tree nodes.|
> |$Z^{(l),k(i)}$|$k(i), l$|Embeddings of node $k$ perturbed by the $i$-th perturbation at layer $l$.|
> |$Z_{\text{final}}^{(l),k}$|$\text{final}, k, l$|Final aggregated embedding for node $k$ at layer $l$, used for routing and decisions.|
>
> Table 6: Feature and Weight-related Symbols.
>
> |**Symbol**|**Subscript/Superscript**|**Meaning and Role**|
> |-|-|-|
> |$X$|None|Input feature matrix, containing the original graph’s node features.|
> |$X^{(l),k}$|$k, l$|Feature matrix of node $k$ at layer $l$, describing its feature state.|
> |$X^{(l),k(i)}$|$k(i), l$|Feature matrix of node $k$ after applying the $i$-th perturbation at layer $l$.|
> |$X^{(l+1)}$|$l+1$|Feature matrix of the coarsened graph at layer $l+1$.|
> |$W^{(l)}$|$l$|Weight matrix of the graph convolution at layer $l$, used for learning graph structural features.|
> |$W^{(1),r,k}$, $W^{(2),r,k}$|$r, k$|Router weight matrices for node $k$ at layer $l$, used to compute path selection probabilities.|
> |$b^{(1),r,k}$, $b^{(2),r,k}$|$r, k$|Bias terms for the router of node $k$ at layer $l$.|
>
> Table 7: Graph Structure-related Symbols.
>
> |**Symbol**|**Subscript/Superscript**|**Meaning and Role**|
> |-|-|-|
> |$A$|None|Adjacency matrix of the original graph.|
> |$\hat{A}$|$\hat{}$| Adjacency matrix with self-loops added.|
> |$A^{(l)}$|$l$| Adjacency matrix of the graph at layer $l$.|
> |$A_{\text{pooled}}^{(l+1),i^{*l,k}}$| $\text{pooled}, i^{*l,k}, l+1$|Adjacency matrix of the coarsened graph generated for the selected path $i^{*l,k}$.|
>
> Table 8: Clustering-related Symbols.
>
> |**Symbol**|**Subscript/Superscript**|**Meaning and Role**|
> |-|-|-|
> |$S^{(l)}$|$l$|Clustering assignment matrix at layer $l$.|
> |$S^{(l),k}$| $k, l$| Clustering assignment matrix for node $k$ at layer $l$.|
> |$S^{(l),k(i)}$| $k(i), l$|Clustering assignment matrix for node $k$ under the $i$-th perturbation at layer $l$.|
>
> Table 9: Loss and Regularization-related Symbols.
>
> |**Symbol**|**Subscript/Superscript**|**Meaning and Role**|
> |-|-|-|
> |$\mathcal{L}_{\text{link}}$|$\text{link}$| Edge prediction loss.|
> |$\mathcal{L}_{\text{similarity}}$|$\text{similarity}$|Similarity regularization.|
> |$\mathcal{L}_{\text{diversity}}$|$\text{diversity}$| Diversity regularization.|
> |$\mathcal{L}_{\text{entropy}}$|$\text{entropy}$| Entropy regularization, encouraging diversity in path selection.|
> |$\mathcal{L}_{\text{CE}}$|$\text{CE}$| Cross-entropy loss, optimizing classification objectives.|
> |$\mathcal{L}_{\text{total}}$|$\text{total}$| Total loss function, combining classification.|
>
> Table 10: Path and Routing-related Symbols.
>
> |**Symbol**|**Subscript/Superscript**|**Meaning and Role**|
> |-|-|-|
> |$r^{(l),k}$|$k, l$|Routing logits for node $k$ at layer $l$.|
> |$p^{(l),k,i}$| $k, i, l$| Path selection probability for node $k$ at layer $l$, representing the likelihood of selecting branch $i$.|
> |$i^{*l,k}$| $l, k$| Optimal path index for node $k$ at layer $l$.|
> |$\text{Path}^{(l),k}$|$k, l$| Path set at layer $l$, describing the paths associated with node $k$.|
>
> Table 11: Parameters and Hyperparameters.
>
> |**Symbol**|**Subscript/Superscript**|**Meaning and Role**|
> |-|-|-|
> |$\lambda_i$| $i$|Weight of the similarity regularization term.|
> |$\mu$| None|Weight of the diversity regularization term.|
> |$\alpha_1, \alpha_2, \alpha_3$ |$1, 2, 3$| Weight coefficients for edge prediction, perturbation, and entropy regularization terms, respectively.|
> |$M$|None|Number of perturbation branches for each node.|
> |$N$|None|Number of nodes in the current layer.|
> |$K^{(l)}$|$l$| Number of clusters at layer $l$.|
> |$L$|None|Total number of layers in the tree.|

---

> ### Author Response · Authors · 2024-11-22
> **Response (2/4)**
>
> >[W2] Effectiveness of Evaluation Metrics.
>
> Thanks. In our initial manuscript, we have provided all widely recognized and commonly used evaluation metrics in existing interpretability research, including explanation accuracy, consistency, and visual comparisons. These evaluation metrics strictly adhere to the standards established in existing works [1-9]. Additionally, considering the tree structure of our model, we have designed several tailored evaluation metrics, such as Path Consistency and Path Importance.
>
> >[W3] & [Q6] Lack of Visual Comparisons Experiments.
>
> Thanks. (1) In the initial manuscript, we have already provided visualization comparison experiments between TIF and GIP (*Page 9-10, Figure 4*).
>
> (2) As suggested, we have additionally included visualization comparison experiments between TIF and the PortGNN model, as well as the GAST model. The results show that compared to GIP, GSAT, and ProtGNN, TIF's capability to span from fine-grained local interactions to coarse-grained global structures provides a more transparent and interpretable decision-making process, elucidating how various levels of graph information contribute to final model predictions. *These additional results have been updated in Figure 11 (Page 19) of the revised manuscript.*
>
> >[W4] Running Efficiency.
>
> Thanks for the valuable comment. As suggested, we have additionally analyzed the theoretical complexity and compared the time consumption of our proposed framework with several interpretable baselines.
>
> (1) We calculated the theoretical complexity of the proposed TIF framework. For time complexity, updating node embeddings requires $O(N \cdot d^2)$ for dense graphs and $O(E \cdot d)$ for sparse graphs, while clustering and perturbation modules add $O(M \cdot N \cdot d \cdot c)$. Overall, the time complexity is $O(N \cdot d^2 + M \cdot N \cdot d \cdot c)$ (dense graphs) or $O(E \cdot d + M \cdot N \cdot d \cdot c)$ (sparse graphs). For space complexity, dense graphs require $O(N^2)$ for adjacency storage, while sparse graphs need $O(E)$. Adding node embeddings, perturbations, and routing parameters results in $O(N^2 + N \cdot d \cdot (1 + M \cdot c))$ for dense graphs or $O(E + N \cdot d \cdot (1 + M \cdot c))$ for sparse graphs. *These clarifications have been updated in Appendix D.2 (Pages 22-23) of the revised manuscript.*
>
> (2) We present the time required to complete the training of each interpretable model. The results of the time consumption in Table 13 show that our method is only slightly less efficient than KerGNN and GIP. Given that our model outperforms KerGNN and GIP in terms of prediction and explanation performance on the vast majority of datasets, we believe that this slight additional time cost is justified. *These additional results have been updated in Appendix D.2 (Pages 22-23, Table 13) of the revised manuscript, as follows:*
>
> Table 13 (Page 23): Time consumption of different methods. The table shows the time required (in seconds) to finish training for each interpretable model on various datasets. “*” indicates the method requires additional pre-training process which takes nearly 72 hours.
>
> |**Methods**|**ENZYMES**|**D\&D** |**COLLAB**|**MUTAG**|**GraphCycle**|**GraphFive**|
> |-|-|-|-|-|-|-|
> |**ProtGNN**|10245.65s|19312.87s|38021.49s|9239.15s|14396.76s|5022.81s|
> |**KerGNN**|384.73s|1313.59s|1927.34s|401.34s|198.45s|458.22s|
> |**&pi;-GNN**|406.18s|966.94s|1747.55s|462.94s|283.74s|429.82s|
> |**GIB**|711.57s|2923.67s|4681.74s|3107.31s|1159.82s|1208.78s|
> |**GSAT**|482.61s|1388.45s|2979.63s|828.19s|568.27s|649.34s|
> |**GIP**|437.51s|1134.20s|2008.77s|452.26s|235.67s|423.87s|
> |**Ours**|433.17s|1109.70s|2251.30s|503.18s|359.69s|488.15s|
>
>
> >[W5] Theoretical Basis of Proposed Algorithms.
>
> Sorry for the confusion. Existing interpretability methods primarily stem from two aspects: human-cognitive logic interpretability [1-9, 12] and mathematical interpretability [10-11].
>
> In this work, we primarily follow the paradigm of human-cognitive logic interpretability. We observe that existing methods often adopt an inflexible approach that captures graph connectivity at a single specific level, whereas real-world graph tasks typically involve relationships at multiple granularities (e.g., relevant interactions in proteins range from functional groups to amino acids and even to protein domains [14-15]). To address this limitation, we propose TIF, which has been extensively validated through a series of quantitative and qualitative experiments designed in strict adherence to the standards established in existing works [1-9], demonstrating the effectiveness of our approach.

---

> ### Author Response · Authors · 2024-11-22
> **Response (3/4)**
>
> >[Q1] Unclear Representation in Fig 1.
>
> Thanks for the valuable comment. We have added the correspondence between the different granularity levels of proteins and the "finer graph," "moderate graph," and "coarser graph" in Figure 1. These interpretability terms of multi-granularity are inspired by the distinct granularity between enzyme and non-enzyme, ranging from **functional groups** via **amino acids** to **protein molecular** level [14-15]. In this context, the "finer graph" refers to the explanation at the functional group level. The "moderate graph" represents the explanation at the amino acid level. The "coarser graph" represents the explanation at the protein molecular level. We previously mentioned this example in the abstract and introduction of the initial manuscript. *These clarifications have been updated in Figure 1 (Page 2) of the revised manuscript.*
>
> >[Q2] Interpretation of Eq (5).
>
> Sorry for the confusion.
>
> (1) $\hat{A} = A + I$ is the adjacency matrix of the original graph with self-loops. $A^{(l+1)}$ represents the coarser graph's adjacency matrix at layer $l+1$.
>
> (2) In Eq. (5), the adjacency matrix $\hat{{A}} = {A} + {I}$ corresponds to the first layer of the graph, where it has the same dimensions $n \times n$ as the original adjacency matrix ${A}$. As the graph undergoes coarsening in subsequent layers, the number of nodes decreases, and the dimensions of the adjacency matrix are reduced accordingly.
>
> (3) Eq. (5) represents a link prediction loss, which is used to ensure that the connectivity of the graph is maintained during the graph aggregation and compression process. The purpose of this loss function is to minimize the difference between the original adjacency matrix and the aggregated adjacency matrix, thus ensuring that the structure and connectivity of the graph are preserved as much as possible during aggregation.
>
> >[Q3] Theoretical Justification of Explanation Accuracy.
>
> Thanks. In our initial manuscript, we followed this used in existing works [1-9], which is a widely recognized practice within the community and has been broadly adopted. To clarify, the explanation accuracy metric includes a prediction consistency formula [13]:
>
> $\text{ACC}_{\text{exp}} = \frac{|f(G) = f(G_s)|}{|T|}$
>
> Here, $f(G)$ and $f(G_s)$ represent the predictions on the original graph $G$ and explanation subgraph $G_s$, respectively. The metric evaluates whether the predictions match, relying on the softmax output to determine the predicted class. Since softmax provides the probability distribution, the highest-probability class is often interpreted as the model's most confident prediction. Thus, existing works [1-9] consistently treat the GNN’s softmax output as a confidence measure for experiments involving explanation accuracy.
>
>
> >[Q4] Graph Kernels' Capability.
>
> Sorry for the confusion. Graph kernels are primarily used in the Consistency metric [9]. The random walk graph kernel measures the similarity between two graphs by counting the number of shared paths, leveraging the path-sharing hypothesis to effectively capture both local and global structural features of graphs. Its theoretical foundation includes the construction of direct product graphs and the mapping properties of kernel methods, making it suitable for assessing the similarity between graphs and patterns in global interaction modeling [4, 9, 16]. The use of graph kernels and the Consistency metric in interpretability research is widely recognized and accepted.
>
> >[Q5] Explanation Experiments on the Mutag Dataset.
>
> Thanks. In the original manuscript, we have conducted explanation experiments on the MUTAG dataset. On this dataset, we performed a series of interpretability experiments, including Explanation Accuracy, Consistency, Path Consistency, and Path Importance. The results are presented in *Table 2, Figure 3 (b), and Figure 3 (c)*. According to the experimental results, compared to all other models listed in the experiment, our model achieves the best performance on the MUTAG dataset in terms of classification accuracy, path consistency, and path importance. Additionally, it achieves the second-best F1 score and explanation accuracy. This demonstrates that our model performs exceptionally well in both classification and explanation tasks on the MUTAG dataset.

---

> ### Author Response · Authors · 2024-11-22
> **Response (4/4)**
>
> ---
>
> Ref:
>
> [1] Zhitao Ying, Dylan Bourgeois, Jiaxuan You, Marinka Zitnik, and Jure Leskovec. GNNExplainer: Generating explanations for graph neural networks. In *NeurIPS (Advances in Neural Information Processing Systems)*, 2019.
>
> [2] Hao Yuan, Haiyang Yu, Jie Wang, Kang Li, and Shuiwang Ji. On explainability of graph neural networks via subgraph explorations. In *ICML (International Conference on Machine Learning)*, 2021.
>
> [3] Hao Yuan, Jiliang Tang, Xia Hu, and Shuiwang Ji. XGNN: Towards model-level explanations of graph neural networks. In *SIGKDD (ACM Knowledge Discovery and Data Mining)*, 2020.
>
> [4] Aosong Feng, Chenyu You, Shiqiang Wang, and Leandros Tassiulas. KerGNNs: Interpretable graph neural networks with graph kernels. In *Proceedings of the AAAI Conference on Artificial Intelligence*, 2022.
>
> [5] Jun Yin, Chaozhuo Li, Hao Yan, Jianxun Lian, and Senzhang Wang. Train Once and Explain Everywhere: Pre-training Interpretable Graph Neural Networks. In *NeurIPS (Advances in Neural Information Processing Systems)*, 2023.
>
> [6] Junchi Yu, Tingyang Xu, Yu Rong, Yatao Bian, Junzhou Huang, and Ran He. Graph Information Bottleneck for Subgraph Recognition. In *ICLR (International Conference on Learning Representations)*, 2020.
>
> [7] Siqi Miao, Mia Liu, and Pan Li. Interpretable and generalizable graph learning via stochastic attention mechanism. In *ICML (International Conference on Machine Learning)*, 2022.
>
> [8] Yongduo Sui, Xiang Wang, Jiancan Wu, Min Lin, Xiangnan He, and Tat-Seng Chua. Causal attention for interpretable and generalizable graph classification. In *SIGKDD (ACM Knowledge Discovery and Data Mining)*, 2022.
>
> [9] Yuwen Wang, Shunyu Liu, Tongya Zheng, Kaixuan Chen, and Mingli Song. Unveiling Global Interactive Patterns across Graphs: Towards Interpretable Graph Neural Networks. In *SIGKDD (ACM Knowledge Discovery and Data Mining)*, pages 3277–3288, 2024.
>
> [10] Sahil Verma, Varich Boonsanong, Minh Hoang, Keegan Hines, John Dickerson, and Chirag Shah. Counterfactual explanations and algorithmic recourses for machine learning: A review. In *ACM Computing Surveys*, volume 56, number 12, pages 1–42, 2024. ACM New York, NY.
>
> [11] Rui Wang, Xiaoqian Wang, and David I. Inouye. Shapley Explanation Networks. In *International Conference on Learning Representations*, 2021.
>
> [12] Hao Yuan, Haiyang Yu, Jie Wang, Kang Li, and Shuiwang Ji. On explainability of graph neural networks via subgraph explorations. In *ICML (International Conference on Machine Learning)*, 2021.
>
> [13] Yiqiao Li, Jianlong Zhou, Sunny Verma, and Fang Chen. A survey of explainable graph neural networks: Taxonomy and evaluation metrics. *arXiv preprint arXiv:2207.12599*, 2022.
>
> [14] Hao Hu, Wei Hu, An-Di Guo, Linhui Zhai, Song Ma, Hui-Jun Nie, Bin-Shan Zhou, Tianxian Liu, Xinglong Jia, Xing Liu, et al. Spatiotemporal and direct capturing global substrates of lysine-modifying enzymes in living cells. *Nature Communications*, **15**(1):1465, 2024.
>
> [15] Yansheng Zhai, Xinyu Zhang, Zijing Chen, Dingyuan Yan, Lin Zhu, Zhe Zhang, Xianghe Wang, Kailu Tian, Yan Huang, Xi Yang, et al. Global profiling of functional histidines in live cells using small-molecule photosensitizer and chemical probe relay labelling. *Nature Chemistry*, pages 1–12, 2024.
>
> [16] SVN Vishwanathan, Karsten M. Borgwardt, Nicol N. Schraudolph, and others. Fast computation of graph kernels. In *NIPS (Advances in Neural Information Processing Systems)*, volume 19, pages 131–138, 2006.

---

> ### Author Response · Authors · 2024-11-25
> **Looking forward to your reevaluation**
>
> Dear reviewer,
>
> We are glad that the reviewer appreciates our attempt, and sincerely thank the reviewer for the constructive comments.
> As suggested, we have additionally added some clarifications about our framework and included further experiments, such as visual comparisons experiments and running efficiency. Please let us know if you have other questions or comments.
>
> Since half of the allocated time for reviewer-author discussion has already elapsed, we sincerely look forward to your reevaluation of our work and would very appreciate it if you could raise your score to boost our chance of more exposure to the community. Thank you very much!
>
> Best regards,
>
> The authors of TIF

---

> > ### Comment · Reviewer_Yj5d · 2024-11-26
> >
> > Hi Authors,
> >
> > After reading the rebuttal, I have decided to raise my score to 5. I agree with Reviewer hGDZ that the correlation between the coarsened graph at each layer and the ground-truth interpretation should be further elaborated. And further, does this approach have broad applicability? For example, in social networks or other networks with simpler structures?
> >
> > Best,
> > Yj5d

---

> > > ### Author Response · Authors · 2024-11-28
> > > **We kindly look forward to your reevaluation**
> > >
> > > Dear Reviewer Yj5d,
> > >
> > > We are truly honored by the reviewer's kind acknowledgment of our efforts, and we sincerely express our gratitude for the insightful and constructive comments provided. In line with your valuable latest suggestions, we have sincerely added further clarifications regarding our framework, such as the correlation between the coarsened graph at each layer and the ground-truth, and the broad applicability of our framework. We would be deeply grateful if you could kindly let us know if there are any further questions or comments.
> > >
> > >
> > > Best regards,
> > >
> > > The authors of TIF

---

> ### Author Response · Authors · 2024-11-26
> **Response**
>
> We are deeply honored to receive your response and your recognition of our work, and we sincerely appreciate your thoughtful feedback.
>
> **[Q1] Correlation of Coarsen Graph to the Ground-truth**
>
> *Reviewer hGDZ has acknowledged our viewpoints and analysis on this aspect and has provided us with positive feedback and a higher score. For further details, we kindly invite you to refer to our discussion with Reviewer hGDZ.* Below, we will take Figure 8 as an example and provide a detailed analysis of the entire process to elaborate correlation between the coarsened graph at each layer and the ground-truth interpretation. Additionally, We have added a visual structure of the ground truth used at each granularity level in the MultipleCycle dataset. *These additional clarifications have been updated on Pages 18-20, Figure 8.*
> + Firstly, the MultipleCycle dataset corresponds to different ground truths at different levels of granularity. As shown in Figure 8, the **input graph** is a sample from the MultipleCycle dataset, and its category is classified as "Hybrid Cycle." Specifically:
>     - Its first-level structure is set as a cycle structure based on the ground truth at this granularity level, which determines its cycle attribute.
>     - Its second-level structure is built on the first-level structure, configured as a mixed combination of cycle and non-cycle structures according to the ground truth at this granularity level. The clockwise sequence is cycle, non-cycle, cycle, and cycle, which determines its mixed attribute.
> + Secondly, we can observe that the **finer graphs** display the second-level structure of the input graph. The model selects the third finer graph, which best reflects the structural information of the input graph (shows a second-level structure proceeding clockwise as cycle, non-cycle, cycle, and cycle). **From the perspective of ground truth**, the model selects the finer graph that is closest to the ground truth structure and layout of the input graph at this granularity level.
> + Subsequently, we can observe that the **coarser graphs** clearly capture the first-level structure of the input graph, which is the cycle structure. The model selects the fourth coarser graph (corresponding to the second-level structure depicted in the finer graph from the previous layer), which best represents the structural information of the input graph, as the root node of the TIF. **From the perspective of ground truth**, the model selects the finer graph that is closest to the ground truth structure and layout of the input graph at this granularity level, while also most accurately preserving the ground truth structural information from the previous granularity level.
> + Finally, at the **root node** of the TIF, the prediction is performed, and the model successfully identifies the category of the data as "Hybrid Cycle." **From an interpretability perspective**, the TIF tree model effectively captures and explains the key attributes of the MultipleCycle dataset at two distinct granularity levels.
>     - The second-level granularity, which characterizes the attributes of being purely cycle, purely non-cycle, or a mixed combination of cycle and non-cycle structures.
>     - The first-level granularity, which identifies whether the structure is cycle or non-cycle.
> + **From the perspective of ground truth**, The relationship between each coarsened graph and the ground truth lies in the fact that each coarsened graph in the TIF model strives to represent the critical structures constructed by the ground truth at the granularity level that the layer aims to explain for the input graph. That is, the coarsened graph obtained at each level by TIF corresponds to the ground truth at that level of granularity.
>  ---
> **[Q2] Broad Applicability**
>
> + Our method demonstrates broad applicability. In the initial manuscript, we conducted experiments on the COLLAB social network dataset, evaluating prediction performance using metrics such as accuracy and F1 score, as well as explanation performance metrics including explanation accuracy, path consistency, and path importance. The experimental results show that, compared to other baselines, our model attained optimal performance across all explanation metrics and the second-best accuracy on the COLLAB dataset.
> + From an interpretability perspective, when the dataset is input as a graph into the model, the model uses the coarsened graph at each layer to explain the relational structures between different scales of social clusters, such as collaborators, groups, teams, and domain communities in the social network. For example, the model uses finer graphs with the scale of "collaborators" as the reference unit to show the social relationships between collaborators. The model uses coarser graphs with the scale of "domain communities" as the reference unit to show the social relationships between domain communities. Finally, the model outputs the results of the classification task.

---

> > ### Comment · Reviewer_Yj5d · 2024-11-28
> >
> > Thanks for your explanation. I have decided to further increase the score.

---

> > > ### Author Response · Authors · 2024-11-28
> > > **Thanks for your positive support**
> > >
> > > We are deeply grateful for your positive support and thoughtful encouragement. Have a nice day！

---

### Official Review · Reviewer_ka23 · 2024-11-04

**Soundness:** 3
**Presentation:** 3
**Contribution:** 3
**Rating:** 6
**Confidence:** 4

**Summary:**

The paper proposes a TIF that transforms GNNs into hierarchical tree structures to provide multi-granular interpretability for graph classification tasks.

**Strengths:**

The paper is very interesting, and the author finds that multi-granular interpretability in GNNs is important but was overlooked by previous works. Thus, they propose TIF to enhance multi-granular interpretability. Extensive experiments show that the proposed method is effective and demonstrates good interpretability.

**Weaknesses:**

Overall the paper is good but why does Table 2 have no std value?

The framework requires significant computational resources to maintain and process multiple branches and paths per node, along with multiple perturbation matrices for each node. The hierarchical tree structure adds computational complexity at each level.

The architecture is complex with three major modules (coarsening, perturbation, routing) and requires careful tuning of 5 different hyperparameters (α1-α5). This makes it more difficult to implement and maintain than simpler interpretability approaches, especially considering the additional logic needed for the dynamic hardening mechanism.


There is no formal mathematical justification for why the multi-granular approach improves interpretability, nor are there guarantees regarding the optimality of the tree structure.

**Questions:**

The training process requires balancing various loss terms, but the paper offers limited guidance on optimizing this process.

---

> ### Author Response · Authors · 2024-11-22
> **Response (1/4)**
>
> We are glad that the reviewer appreciates our work as a useful contribution to the community. We have carefully revised the manuscript according to your constructive suggestions. Below we address the main points raised in the review.
>
> >[W1] Std Value.
>
> Sorry for the confusion. Due to formatting and space constraints, we omitted the std values in some tables in the initial manuscript. We have now provided the std values for all experimental data. *The std values of Table 1 have been provide in Table 12 (Page 22, Appendix D.1) of the revised manuscript. The std values of Table 2 have been provide in the original Table 2. All revised tables are presented as follows:*
>
> Table 12 (Page 22): Comparison of different methods in terms of classification accuracy (%) and F1 score (%) along with their corresponding stds.
>
> | Method   | ENZYMES (Acc.) | ENZYMES (F1) | D&D (Acc.) | D&D (F1) | PROTEINS (Acc.) | PROTEINS (F1) | MUTAG (Acc.) | MUTAG (F1) | COLLAB (Acc.) | COLLAB (F1) | GraphCycle (Acc.) | GraphCycle (F1) | GraphFive (Acc.) | GraphFive (F1) | MultipleCycle (Acc.) | MultipleCycle (F1) |
> |-|-|-|-|-|-|-|-|-|-|-|-|-|-|-|-|-|
> |GCN|57.23±0.81|51.32±0.33|76.15±2.77|69.12±1.02|78.89±0.90| 72.21±3.14|71.82±4.27|63.18±4.36|72.56±1.09|65.78±3.75|79.45±1.04|71.56±1.02|57.37±0.81|53.44±0.55|59.64±4.70|55.56±3.34|
> |DGCNN|59.12±3.30|54.89±1.88|78.23±0.78|71.76±1.59|75.36±1.92|71.43±3.53|58.67±0.80|49.21±0.42|74.88±2.38|68.22±1.44|81.12±2.72|75.34±3.11|57.29±3.33|54.43±2.87|60.71±1.07|56.33±2.41|
> |Diffpool|61.01±2.26|56.98±2.55|81.56±1.31|75.43±4.68|79.52±0.78|78.22±0.87|84.12±2.18|72.45±2.60|72.89±1.39|70.12±1.17|78.34±4.23|71.87±4.59|55.46±1.21|53.57±1.57|56.87±2.03|53.21±2.22|
> |RWNN|54.76±1.43|48.12±3.22|76.89±1.99|74.67±2.18|76.12±1.36|70.89±1.40|88.21±0.21|85.04±0.41|73.45±1.52|68.45±1.97|78.89±1.48|78.76±2.53|56.25±0.42|52.45±1.22|57.16±5.56|54.09±4.10|
> |GraphSAGE|58.12±1.22|44.89±1.32|79.34±5.31|79.23±6.77|79.04±2.15|68.45±2.08|74.23±3.27|71.78±3.62|71.23±1.58|65.45±2.51|77.45±1.49|72.12±2.47|59.11±0.34|52.72±0.36|62.66±0.21|59.34±0.77|
> |ProtGNN|53.21±1.57|43.89±2.36|76.12±1.21|75.23±2.49|76.89±0.52|72.45±1.87|80.34±2.45|61.23±3.83|70.12±0.97|67.89±1.04|80.12±1.21|72.34±2.04|56.38±4.21|54.32±4.37|60.26±3.38|58.41±3.67|
> |KerGNN|55.67±4.22|48.45±2.03|72.89±1.48|68.23±2.36|76.12±2.30|71.12±2.10|71.45±1.08|62.12±1.22|74.12±1.66|69.12±1.97|80.21±0.72|73.89±0.68|58.06±0.11|50.82±1.02|63.22±0.05|57.94±0.33|
> |&pi;-GNN|55.34±0.88|47.12±0.76|79.12±1.10|73.89±1.85|72.34±3.77|68.12±2.21|90.12±0.43|75.12±2.09|73.45±1.52|68.34±3.05|81.45±2.22|76.78±5.62|60.14±0.05|54.07±0.31|64.74±1.21|62.48±1.97|
> |GIB|45.12±3.22|31.67±1.73|77.34±1.69|66.45±0.90|75.12±6.34|70.34±1.05|91.03±4.88|82.12±1.26|73.34±1.79|61.89±1.65|80.67±1.74|74.12±1.98|59.78±0.15|59.24±0.17|63.23±2.63|63.02±2.70|
> |GSAT|61.34±0.65|55.12±1.47|72.12±1.13|67.12±3.22|74.45±0.79|71.89±1.48|94.35±1.12|82.34±1.93|75.87±3.56|63.78±2.59|80.12±0.14|75.08±0.57|59.58±3.09|54.13±2.70|66.49±1.50|65.24±1.53|
> |CAL|61.12±3.24|58.12±4.44|78.12±2.88|68.78±4.76|74.56±4.09|67.12±4.21|89.78±6.99|85.12±8.31|77.12±4.78|64.12±6.25|81.42±2.33|78.12±2.40|56.49±1.44|50.93±2.59|61.77±0.42|58.94±1.73|
> |GIP|60.61±2.41|57.41±2.80|79.32±1.01|75.78±0.36|79.55±0.61|75.28±0.90|91.21±2.25|86.73±2.92|77.49±4.26|67.47±2.11|82.15±1.38|78.31±2.66|60.38±3.33|54.98±1.52|68.72±0.02|66.45±1.34|
> |Ours|58.66±1.44|55.44±2.50|84.19±0.88|81.01±0.76|79.96±0.97|77.21±0.34|94.44±2.44|86.23±3.52|77.29±2.08|67.82±3.27|84.77±0.92|78.49±1.16|64.35±3.55|55.07±2.87|69.04±0.21|67.91±2.77|
>
> Table 2 (Page 9): Comparison of different methods in terms of explanation accuracy.
>
> | **Method**      | **ENZYMES**        | **D&D**            | **PROTEINS**       | **MUTAG**          | **COLLAB**         | **GraphCycle**     | **GraphFive**      | **MultipleCycle**  |
> |-|-|-|-|-|-|-|-|-|
> | **ProtGNN**|85.12±2.12|80.34±2.45|69.12±3.12|71.23±2.56|80.67±2.78|81.23±1.56|72.12±1.34|74.86±2.15|
> | **KerGNN**|63.45±2.45|60.78±2.12|79.12±1.45|88.12±0.78|84.11±1.12|85.78±0.34|75.67±0.67|76.41±1.22|
> | **π-GNN**|75.12±1.12|81.34±0.34|65.45±2.67|81.30±4.12|76.12±0.34|83.45±1.45|64.89±0.34|70.51±3.28|
> | **GIB**|72.89±2.12|76.34±2.78|82.78±1.67|85.12±2.45|78.45±2.34|86.08±1.01|79.07±0.56|80.74±2.54|
> | **GSAT**|82.45±2.67|75.12±0.45|60.34±1.23|75.34±3.56|75.89±2.78|90.12±2.34|59.12±1.01|68.38±1.79|
> | **CAL**|77.78±1.34|74.12±3.12|64.12±2.45|76.12±1.37|84.78±1.22|84.12±2.12|82.48±2.33|84.12±2.12|
> | **GNNExplainer**|80.12±0.45|79.45±2.12|87.12±2.45|82.13±2.12|71.34±3.12|85.41±2.78|70.12±2.45|77.34±2.05|
> | **SubgraphX**|81.67±2.45|71.23±1.01|75.89±2.12|87.45±3.12|76.34±3.12|91.12±2.01|69.50±3.45|80.44±2.06|
> | **XGNN**|87.34±2.12|74.45±2.56|74.12±2.01|83.12±4.01|84.45±0.56|86.12±0.34|76.45±1.17|84.37±3.81|
> | **GIP**|86.08±2.60|83.47±2.74|86.04±2.36|90.05±1.44|85.21±3.72|92.79±1.32|78.76±1.57|85.16±2.68|
> | **Ours**|86.53±2.01|89.11±1.26|87.62±2.12|88.21±1.34|85.95±3.64|93.12±1.12|82.16±1.33|86.95±2.70|

---

> ### Author Response · Authors · 2024-11-22
> **Response (2/4)**
>
> >[W2] Running Efficiency.
>
> Thanks. As suggested, we have additionally analyzed the theoretical complexity and compared the time consumption of our proposed framework with several interpretable baselines.
>
> (1) We calculated the theoretical complexity of the proposed TIF framework. For time complexity, updating node embeddings requires $O(N \cdot d^2)$ for dense graphs and $O(E \cdot d)$ for sparse graphs, while clustering and perturbation modules add $O(M \cdot N \cdot d \cdot c)$. Overall, the time complexity is $O(N \cdot d^2 + M \cdot N \cdot d \cdot c)$ (dense graphs) or $O(E \cdot d + M \cdot N \cdot d \cdot c)$ (sparse graphs). For space complexity, dense graphs require $O(N^2)$ for adjacency storage, while sparse graphs need $O(E)$. Adding node embeddings, perturbations, and routing parameters results in $O(N^2 + N \cdot d \cdot (1 + M \cdot c))$ for dense graphs or $O(E + N \cdot d \cdot (1 + M \cdot c))$ for sparse graphs. *These clarifications have been updated in Appendix D.2 (Pages 22-23) of the revised manuscript.*
>
> (2) We present the time required to complete the training of each interpretable model. The results of the time consumption in Table 13 show that our method is only slightly less efficient than KerGNN and GIP. Given that our model outperforms KerGNN and GIP in terms of prediction and explanation performance on the vast majority of datasets, we believe that this slight additional time cost is justified. *These additional results have been updated in Appendix D.2 (Pages 22-23, Table 13) of the revised manuscript, as follows:*
>
> Table 13 (Page 23): Time consumption of different methods. The table shows the time required (in seconds) to finish training for each interpretable model on various datasets. “*” indicates the method requires an additional pre-training process which takes nearly 72 hours.
>
> |**Methods**|**ENZYMES**|**D\&D** |**COLLAB**|**MUTAG**|**GraphCycle**|**GraphFive**|
> |-|-|-|-|-|-|-|
> |**ProtGNN**|10245.65s|19312.87s|38021.49s|9239.15s|14396.76s|5022.81s|
> |**KerGNN**|384.73s|1313.59s|1927.34s|401.34s|198.45s|458.22s|
> |**&pi;-GNN**|406.18s|966.94s|1747.55s|462.94s|283.74s|429.82s|
> |**GIB**|711.57s|2923.67s|4681.74s|3107.31s|1159.82s|1208.78s|
> |**GSAT**|482.61s|1388.45s|2979.63s|828.19s|568.27s|649.34s|
> |**GIP**|437.51s|1134.20s|2008.77s|452.26s|235.67s|423.87s|
> |**Ours**|433.17s|1109.70s|2251.30s|503.18s|359.69s|488.15s|

---

> ### Author Response · Authors · 2024-11-22
> **Response (3/4)**
>
> >[W3] & [Q1] (1) Excessive Hyperparameters, (2) Dynamic Hardening Mechanism Increases the Complexity of the Method.
>
> Sorry for the confusion. (1) The initial manuscript contained errors in the number and values of the hyperparameters, which we have now corrected. Our model uses only three parameters (α1–α3) across different datasets, and their values remain consistent without any additional adjustments (we have provided our code in the original supplementary material, as detailed in lines 654, 445, and 620 of encoders.py). This demonstrates the robustness of our method, which does not require hyperparameter tuning. *These clarifications have been updated in Table 4 (Page 16) of the revised manuscript, as follows:*
>
> Table 4 (Page 16): The statistics of hyper-parameters setting.
>
> | |**ENZYMES**|**PROTEINS**|**D\&D**|**MUTAG**|**COLLAB**|**GraphCycle**|**GraphFive**|**MultipleGraph**|
> |-|-|-|-|-|-|-|-|-|
> |**Batch Size**|64|64|128|64|64|128|128|128|
> |**Optimizer**|Adam|Adam|Adam|Adam|Adam|Adam|Adam|Adam|
> |**Learning Rate**|0.001|0.003|0.001|0.001|0.003|0.01|0.01|0.01|
> |**Epoch**|500|500|500|500|500|500|500|500|
> |**$α_1 / α_2$**|0.3/0.2|0.3/0.2|0.3/0.2|0.3/0.2|0.3/0.2|0.3/0.2|0.3/0.2|0.3/0.2|
> |**$α_3$**|0.1|0.1|0.1|0.1|0.1|0.1|0.1|0.1|
>
> (2) We initially designed the dynamic hardening mechanism and included it in the appendix of the initial manuscript to explore better interpretability. However, during subsequent experiments, we found that it did not achieve the desired effect. None of the experiments used this module, but we forgot to remove it from the appendix. *In the revised manuscript, we have removed this content from the appendix.*
>
> >[W4] (1) Lack of Mathematical Proof for Approach Effectiveness， (2) the Optimality of the Tree Structure.
>
> (1) Sorry for the confusion. Existing interpretability methods primarily stem from two aspects: human-cognitive logic interpretability [3-12] and mathematical interpretability [1-2].
>
> In this work, we primarily follow the paradigm of human-cognitive logic interpretability. We observe that existing methods often adopt an inflexible approach that captures graph connectivity at a single specific level, whereas real-world graph tasks typically involve relationships at multiple granularities (e.g., relevant interactions in proteins range from functional groups to amino acids and even to protein domains [13-14]). To address this limitation, we propose TIF, which has been extensively validated through a series of quantitative and qualitative experiments designed in strict adherence to the standards established in existing works [3-12], demonstrating the effectiveness of our approach.
>
> (2) Sorry for the misunderstanding. In the initial manuscript of the main text (*Pages 9-10*) and Appendix C (*Pages 20-22*), we provided a series of ablation studies. These studies included analyses from perspectives such as compression ratio, path count, and routing structure. Based on the experimental results, it can be concluded that the tree structure currently used by TIF is indeed the optimal tree structure.

---

> ### Author Response · Authors · 2024-11-22
> **Response (4/4)**
>
> ---
>
> Ref:
>
> [1] Sahil Verma, Varich Boonsanong, Minh Hoang, Keegan Hines, John Dickerson, and Chirag Shah. Counterfactual explanations and algorithmic recourses for machine learning: A review. In *ACM Computing Surveys*, volume 56, number 12, pages 1–42, 2024. ACM New York, NY.
>
> [2] Rui Wang, Xiaoqian Wang, and David I. Inouye. Shapley Explanation Networks. In *International Conference on Learning Representations*, 2021.
>
> [3] Zhitao Ying, Dylan Bourgeois, Jiaxuan You, Marinka Zitnik, and Jure Leskovec. GNNExplainer: Generating explanations for graph neural networks. In *NeurIPS (Advances in Neural Information Processing Systems)*, 2019.
>
> [4] Hao Yuan, Haiyang Yu, Jie Wang, Kang Li, and Shuiwang Ji. On explainability of graph neural networks via subgraph explorations. In *ICML (International Conference on Machine Learning)*, 2021.
>
> [5] Hao Yuan, Jiliang Tang, Xia Hu, and Shuiwang Ji. XGNN: Towards model-level explanations of graph neural networks. In *SIGKDD (ACM Knowledge Discovery and Data Mining)*, 2020.
>
> [6] Zaixi Zhang, Qi Liu, Hao Wang, Chengqiang Lu, and Cheekong Lee. ProtGNN: Towards self-explaining graph neural networks. In *AAAI (Association for the Advancement of Artificial Intelligence)*, 2022.
>
> [7] Aosong Feng, Chenyu You, Shiqiang Wang, and Leandros Tassiulas. KerGNNs: Interpretable graph neural networks with graph kernels. In *Proceedings of the AAAI Conference on Artificial Intelligence*, 2022.
>
> [8] Jun Yin, Chaozhuo Li, Hao Yan, Jianxun Lian, and Senzhang Wang. Train Once and Explain Everywhere: Pre-training Interpretable Graph Neural Networks. In *NeurIPS (Advances in Neural Information Processing Systems)*, 2023.
>
> [9] Junchi Yu, Tingyang Xu, Yu Rong, Yatao Bian, Junzhou Huang, and Ran He. Graph Information Bottleneck for Subgraph Recognition. In *ICLR (International Conference on Learning Representations)*, 2020.
>
> [10] Siqi Miao, Mia Liu, and Pan Li. Interpretable and generalizable graph learning via stochastic attention mechanism. In *ICML (International Conference on Machine Learning)*, 2022.
>
> [11] Yongduo Sui, Xiang Wang, Jiancan Wu, Min Lin, Xiangnan He, and Tat-Seng Chua. Causal attention for interpretable and generalizable graph classification. In *SIGKDD (ACM Knowledge Discovery and Data Mining)*, 2022.
>
> [12] Yuwen Wang, Shunyu Liu, Tongya Zheng, Kaixuan Chen, and Mingli Song. Unveiling Global Interactive Patterns across Graphs: Towards Interpretable Graph Neural Networks. In *SIGKDD (ACM Knowledge Discovery and Data Mining)*, pages 3277–3288, 2024.
>
> [13] Hao Hu, Wei Hu, An-Di Guo, Linhui Zhai, Song Ma, Hui-Jun Nie, Bin-Shan Zhou, Tianxian Liu, Xinglong Jia, Xing Liu, et al. Spatiotemporal and direct capturing global substrates of lysine-modifying enzymes in living cells. *Nature Communications*, 15(1):1465, 2024.
>
> [14] Yansheng Zhai, Xinyu Zhang, Zijing Chen, Dingyuan Yan, Lin Zhu, Zhe Zhang, Xianghe Wang, Kailu Tian, Yan Huang, Xi Yang, et al. Global profiling of functional histidines in live cells using small-molecule photosensitizer and chemical probe relay labelling. *Nature Chemistry*, pages 1–12, 2024.

---

> ### Author Response · Authors · 2024-11-25
> **Looking forward to your reevaluation**
>
> Dear reviewer,
>
> We are glad that the reviewer appreciates our attempt, and sincerely thank the reviewer for the constructive comments. As suggested, we have additionally added some clarifications about our framework and included further experiments, such as those on running efficiency. Please let us know if you have other questions or comments.
>
> Since half of the allocated time for reviewer-author discussion has already elapsed, we sincerely look forward to your reevaluation of our work and would very appreciate it if you could raise your score to boost our chance of more exposure to the community. Thank you very much!
>
> Best regards,
>
> The authors of TIF

---

> > ### Comment · Reviewer_ka23 · 2024-11-26
> >
> > Thanks for your response. It is better to add these new results to your paper.

---

> > > ### Author Response · Authors · 2024-11-27
> > > **Thanks for your positive support**
> > >
> > > We are deeply grateful for your positive support and thoughtful encouragement.  We have added these new results to the paper, and the std values of some results have been updated in the new table in the appendix. Have a nice day！

---

### Meta-Review · Area_Chair_3ph5 · 2024-12-18

**Metareview:**

### Summary
The paper proposes a Tree-like Interpretable Framework (TIF) to improve interpretability in graph neural networks (GNNs) for graph classification tasks. TIF transforms GNNs into hierarchical tree structures, where each layer represents different graph granularities. It includes three major components: Graph Coarsening Module (Compresses graphs into coarser representations while preserving diversity), Graph Perturbation Module (Introduces perturbations to enhance robustness and diversity), and Adaptive Routing Module (Selects the most informative root-to-leaf path for interpretation and prediction).
Experiments show that TIF improves multi-granular interpretability while maintaining competitive predictive performance compared to state-of-the-art interpretable GNNs.

### Strengths
- The hierarchical tree structure provides a multi-granular approach to graph interpretability, addressing an overlooked need in prior works.
- The paper conducts extensive experiments with comparisons against multiple baseline methods, demonstrating the effectiveness of TIF in both prediction and interpretation.
- The manuscript is well-structured, and the figures and tables are insightful, making the proposed framework easy to understand.

### Weaknesses
- The hierarchical tree structure and multi-module design significantly increase computational cost. Multiple hyperparameters and perturbation matrices per node add complexity, making implementation and maintenance challenging.
- The framework relies heavily on heuristics without clear mathematical justification for why the multi-granular approach improves interpretability. The optimality of the tree structure is not guaranteed.
- Some evaluation metrics fail to reflect the sparsity or quality of explanations, limiting their practical relevance. Key definitions (e.g., interpretation, path importance, compression ratio) are missing or introduced late, causing confusion. Figures and comparisons lack sufficient detail (e.g., how TIF explanations compare to ground-truth or baseline methods).

The paper proposes an innovative and practical framework for improving interpretability in GNNs, addressing a significant gap in graph classification tasks. The multi-granular interpretability provided by the hierarchical tree structure is a valuable contribution, and the extensive experiments demonstrate the potential of TIF to advance interpretable GNNs. While theoretical justifications and computational efficiency could be improved, the novelty and practical implications of the framework justify its acceptance.

**Additional Comments On Reviewer Discussion:**

Major concerns raised by the reviewers are summarized in the above weaknesses. The authors addressed most of the major concerns and satisfied the reviewers.

---

### Decision · Program_Chairs · 2025-01-22

Accept (Poster)